# DuoAttention: Efficient Long-Context LLM Inference with Retrieval and Streaming Heads

**Guangxuan Xiao**[1] *    **Jiaming Tang**[1]    **Jingwei Zuo**[2]    **Junxian Guo**[1,3]
**Shang Yang**[1]    **Haotian Tang**[1]    **Yao Fu**[4]    **Song Han**[1,5]
[1] MIT    [2] Tsinghua University    [3] SJTU    [4] University of Edinburgh    [5] NVIDIA
`https://github.com/mit-han-lab/duo-attention`

## Abstract

Deploying long-context large language models (LLMs) is essential but poses significant computational and memory challenges. Caching all Key and Value (KV) states across all attention heads consumes substantial memory. Existing KV cache pruning methods either damage the long-context capabilities of LLMs or offer only limited efficiency improvements. In this paper, we identify that only a fraction of attention heads, a.k.a, *Retrieval Heads*, are critical for processing long contexts and require full attention across all tokens. In contrast, all other heads, which primarily focus on recent tokens and attention sinks–referred to as *Streaming Heads*–do not require full attention. Based on this insight, we introduce DuoAttention, a framework that only applies a full KV cache to retrieval heads while using a light-weight, constant-length KV cache for streaming heads, which reduces both LLM's decoding and pre-filling memory and latency without compromising its long-context abilities. DuoAttention uses a lightweight, optimization-based algorithm with synthetic data to identify retrieval heads accurately. Our method significantly reduces long-context inference memory by up to $2.55\times$ for MHA and $1.67\times$ for GQA models while speeding up decoding by up to $2.18\times$ and $1.50\times$ and accelerating pre-filling by up to $1.73\times$ and $1.63\times$ for MHA and GQA models, respectively, with minimal accuracy loss compared to full attention. Notably, combined with quantization, DuoAttention enables Llama-3-8B decoding with 3.3 million context length on a single A100 GPU. Code is provided in the link.

## 1 Introduction

Large language models (LLMs) (Touvron et al., 2023a;b; OpenAI, 2023; Black et al., 2022) are at the forefront of the AI revolution, powering advanced applications such as multi-round dialogues (Schulman et al., 2022; Taori et al., 2023; Chiang et al., 2023), long document summarization (Goyal & Durrett, 2020; Zhang et al., 2023a), and tasks involving mixed modalities like visual and video understanding (Liu et al., 2023b; Lin et al., 2023). These applications often require processing extensive numbers of contextual tokens; for instance, summarizing the entire Harry Potter series could involve analyzing approximately one million tokens. The challenge intensifies with visual language models (VLMs), where a single $224\times224$ image corresponds to 256 tokens (Liu et al., 2023b), and a three-minute video at 24 FPS generates around 1.1 million tokens.

A critical issue in deploying LLMs in such applications is the long-context inference problem. The full attention mechanism demands that all tokens attend to every previous token for accurate representation, resulting in linearly increasing decoding and quadratically increasing pre-filling latency as the sequence length grows. Additionally, the Key-Value (KV) Cache technique, which stores keys and values from all preceding tokens, causes memory usage to scale linearly with context length. As sequences lengthen, memory is increasingly consumed by the KV cache, placing a significant computational burden on the attention mechanism. For instance, in the Llama-3-8B (Dubey et al., 2024) model architecture, serving with FP16 KV cache for 1 million tokens would require at least 137 GB of memory—exceeding the capacity of a single 80GB GPU. Additionally, the latencies

---

*Part of the work done during an internship at NVIDIA.

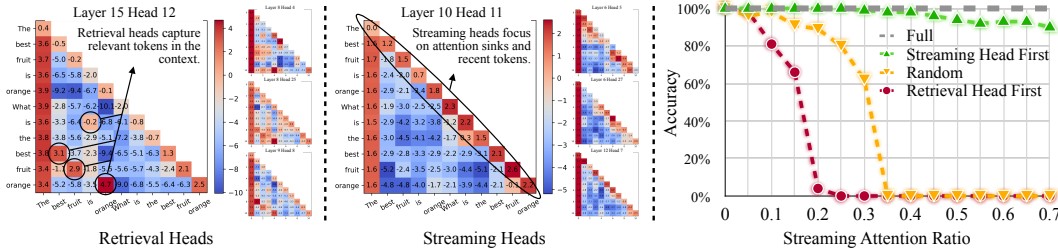

Figure 1: Visualization of attention maps in the Llama-2-7B model for the sentence "*The best fruit is orange. What is the best fruit? Orange.*" shows the distinct roles of *retrieval heads* (e.g., Layer 15, Head 12) and *streaming heads* (e.g., Layer 10, Head 11). On the left, retrieval heads capture contextually relevant tokens such as "best," "fruit," and "orange," which are crucial for processing long-context information and, therefore, require a full KV cache. In the middle, streaming heads primarily focus on initial and recent tokens without emphasizing past contextual relevance. On the right, the impact of limiting attention to the sink and recent tokens on long-context passkey retrieval accuracy is shown: modifying retrieval heads severely damages performance, while constraining streaming heads has minimal impacts.

of pre-filling and decoding with such large contexts are significant, posing substantial challenges to the effective use of LLMs in long-context scenarios.

Despite numerous efforts to overcome the challenges of attention mechanisms in long-context inference, significant computational and memory issues persist. Architectural modifications, such as Grouped-Query Attention (GQA)(Ainslie et al., 2023), require model pre-training and fail to reduce computational costs. Linear Attention methods (Gu & Dao, 2023; Poli et al., 2023), while less demanding in terms of computation and memory, often underperform in long-context scenarios compared to Transformer models. Approximative attention methods, such as $H_2O$ (Zhang et al., 2023b), StreamingLLM (Xiao et al., 2023b), TOVA (Oren et al., 2024), and FastGen (Ge et al., 2024), often compromise accuracy in long-context applications. KV cache quantization (Liu et al., 2024; Hooper et al., 2024), although useful, does not reduce the computation time of the attention mechanism. System-level optimizations, including FlashAttention (Dao et al., 2022; Dao, 2023), FlashDecoding (Hong et al., 2024), and PagedAttention (Kwon et al., 2023), while effective, do not reduce the KV cache size and still require significant computation for extended contexts. These limitations emphasize the need for further advancements to deploy models that handle million-level context lengths.

In this paper, we introduce a key observation that attention heads in LLMs can be categorized into two distinct types: Retrieval Heads (Wu et al., 2024) and Streaming Heads, as shown in Figure 1. *Retrieval Heads*, which represent only a fraction of the total, are crucial for processing long contexts and require full attention across all tokens. In contrast, the majority of attention heads, termed *Streaming Heads*, primarily focus on recent tokens and attention sinks (Xiao et al., 2023b), and can operate effectively with a reduced KV cache that includes only recent tokens and attention sinks.

Building on the dichotomy of retrieval and streaming heads, we propose DuoAttention, a general, straightforward, and easily integrated approach that significantly accelerates both LLM's decoding and pre-filling and reduces memory footprints, particularly in long-context scenarios. The core innovation of DuoAttention is a lightweight, optimization-based procedure that identifies non-compressible retrieval heads using synthetic datasets. Unlike existing methods that rely on attention pattern profiling (Wu et al., 2024; Ge et al., 2024; Tang et al., 2024a), DuoAttention directly measures output deviation resulting from token dropping, achieving higher compression rates and improved deployment efficiency. DuoAttention is designed with simplicity and efficiency in mind: each Transformer layer has two KV caches— a full KV cache for crucial retrieval heads and a constant KV cache for streaming heads, which stores only attention sinks and recent tokens. This design allows DuoAttention to dramatically reduce memory usage and improve decoding speed in models like Llama-2/3 and Mistral, achieving up to 2.55× for MHA and 1.67× for GQA models while speeding up decoding by up to 2.18× and 1.50× and accelerating pre-filling by up to 1.73× and 1.63× for MHA and GQA models, respectively, with minimal accuracy loss compared to full attention.

Moreover, DuoAttention is fully compatible with important optimization techniques like GQA and quantization. We show that when combined with 8-bit weight 4-bit KV cache quantization, DuoAttention enables a Llama-3-8B model to handle up to 3.3 million contextual tokens measured on a single A100 GPU, achieving a 6.4× capacity increase compared to standard full attention FP16 deployments. DuoAttention paves the way for deploying LLMs in applications requiring million-level context handling.

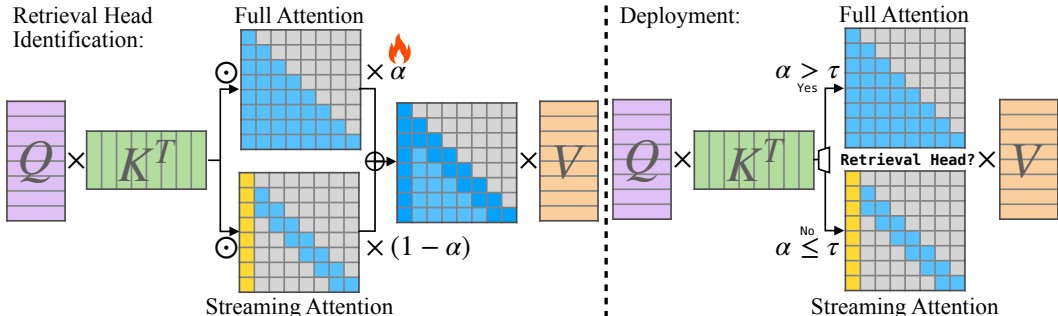

Figure 2: **Overview of DuoAttention:** (1) In the retrieval head identification phase, we assign a trainable gate value, $\alpha$, to each attention head, which blends the outputs of full attention and streaming attention. The training objective is to optimize these values to minimize the deviation from the full attention model's output, while simultaneously applying a regularization loss to encourage lower gate values. This training phase is efficient, requiring only the gate values to be trainable—leaving all other model parameters frozen—thus allowing it to be completed within several hours on an 8 GPU node. (2) During deployment, these gate values are binarized to classify heads as either retrieval or streaming based on a threshold $\tau$. Retrieval heads, identified by a gate value above the threshold, use full attention, caching the KV pairs for all tokens. In contrast, streaming heads cache only the KV pairs of recent tokens and attention sinks.

## 2 DUOATTENTION

### 2.1 RETRIEVAL AND STREAMING HEADS

**Retrieval Heads**  In Transformer-based LLMs, attention heads exhibit distinct and consistent patterns, reflecting their specialized functionalities (Clark et al., 2019; Xiao et al., 2023b; Wu et al., 2024). Figure 1 visualizes two types of attention heads in the Llama-2-7B-32K-Instruct model using the sentence "*The best fruit is orange. What is the best fruit? Orange*". The left panel highlights an attention head that emphasizes relevant tokens during decoding; for instance, the first occurrence of "best fruit" is accentuated while decoding the second "best fruit," and the initial "orange" is highlighted when inferring the second "orange." These attention heads, which we term *Retrieval Heads*, are crucial for context processing as they capture contextually relevant tokens. Compressing the KV cache for retrieval heads would lead to the loss of vital contextual information, and thus they require full attention across all tokens.

**Streaming Heads**  In contrast, the attention head depicted in the middle panel of Figure 1 primarily attends to recent tokens and attention sinks (Xiao et al., 2023b), without highlighting earlier relevant tokens in the context. We refer to these as *Streaming Heads*. Compressing the KV cache for Streaming Heads is feasible because dropping the unattended middle tokens does not significantly alter the attention output. Therefore, streaming heads can be optimized by retaining only the KV states of attention sinks and recent tokens, without compromising the model's ability to manage long contexts.

**Impact of Token Pruning on Retrieval and Streaming Heads**  The right panel of Figure 1 shows a preliminary passkey retrieval experiment, showing that the model's performance drops significantly when the middle tokens in the KV cache of retrieval heads are pruned, i.e., replaced with streaming attention. In contrast, removing the middle tokens for streaming heads has no significant impact on passkey retrieval accuracy. This observation indicates that we can enhance computational efficiency without sacrificing the model's long-context capabilities: By dropping middle tokens for streaming heads while keeping full attention for retrieval heads, we reduce the memory demands of streaming heads to $O(1)$, thereby improving the efficiency of processing long contexts.

### 2.2 OPTIMIZATION-BASED IDENTIFICATION OF RETRIEVAL HEADS

**Definition of Retrieval Heads**  Section 2.1 qualitatively defines retrieval and streaming heads, but for precise identification, we need a concrete and quantitative definition. In this paper, we define "retrieval heads" as the attention heads that:

*significantly alter model outputs when restricted to recent tokens and attention sinks.*

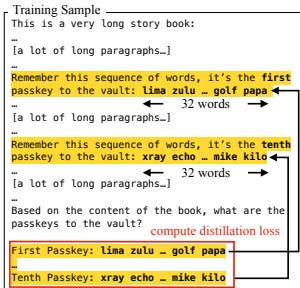

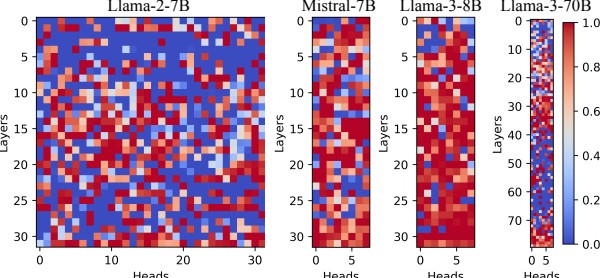

Figure 3: Example from the synthetic dataset used to identify retrieval heads. We embed ten 32-word passkeys within a long text and ask the model to recall these passkeys. Distillation loss is calculated solely on the passkeys.

Figure 4: Optimized gate values of four LLMs. Llama-2-7B uses MHA with 32 heads per layer, while Mistral and Llama-3 models use GQA with 8 heads per layer. Retrieval heads have higher scores. MHA models have a lower ratio of retrieval heads compared to GQA models.

We use this criterion to distinguish retrieval heads from streaming heads. Note that this definition differs from existing works (Ge et al., 2024; Wu et al., 2024; Tang et al., 2024a) that rely solely on attention scores to identify retrieval heads, which overlook 1) the end-to-end impact of compressing the KV cache for specific attention heads, 2) the role of value states, and 3) the variability of attention distributions across layers and heads. In contrast, our definition directly measures output deviation, allowing us to identify attention heads crucial for long-context processing, *even when they are not apparent in attention scores*. We support this argument with ablation studies presented in Section 3.5.

**Optimization-based Identification**  We employ an optimization-based approach to identify retrieval heads, drawing inspiration from prior work in CNN filter pruning (Liu et al., 2017), as illustrated in Figure 2. First, we assign a gate value $\alpha_{i,j}$, to each key-value (KV) head in the LLM. This value intuitively represents the importance of the $j$-th KV head in layer $i$ for processing long-context information. Note that in models using GQA, one KV head can be associated with multiple attention heads, and our method accounts for the KV cache compression of an entire group of attention heads.

Our optimization-based identification method directly assesses the impact of compressing the KV cache with only sink and recent tokens for each KV head. We begin by initializing the gate value $\alpha_{i,j} \in [0, 1]$ for each head at 1, assuming that all heads initially serve as retrieval heads. These gate values are then optimized, with the LLM's parameters remaining fixed, limiting the number of trainable parameters to #layers × #heads and preventing the impact to the model's abilities.

During the forward pass, we combine the outputs of full and streaming attention (which attends only to sink and recent tokens) for each KV head, using the gate value as the mixing weight:

$$\texttt{attn}_{i,j} = \alpha_{i,j} \cdot \texttt{full\_attn} + (1 - \alpha_{i,j}) \cdot \texttt{streaming\_attn}$$

where the attention calculations are defined as:

$$\texttt{full\_attn} = \text{softmax}(\boldsymbol{Q}\boldsymbol{K}^T \odot \boldsymbol{M}_{\text{causal}})\boldsymbol{V},$$

$$\texttt{streaming\_attn} = \text{softmax}(\boldsymbol{Q}\boldsymbol{K}^T \odot \boldsymbol{M}_{\text{streaming}})\boldsymbol{V},$$

where $\boldsymbol{M}_{\text{causal}}$ is the causal attention mask (a lower triangular matrix), and $\boldsymbol{M}_{\text{streaming}}$ represents a $\Lambda$-like mask (Han et al., 2023; Xiao et al., 2023b) that attends only to recent and initial tokens.

**Synthetic Dataset for Identifying Retrieval Heads**  However, relying solely on natural language modeling objectives is insufficient for identifying retrieval heads because the supervision signal in natural text that requires inference over long spans is sparse, and most tokens can be inferred using local context. To address this, we design a synthetic dataset specifically aimed at enhancing the model's long-context retrieval capabilities, allowing us to effectively identify which KV heads can be compressed without compromising the model's performance. As depicted in Figure 3, we create a passkey-retrieval dataset by embedding ten randomly generated passkey sequences of $s$ tokens in ten random locations within a very long context ($s = 32$ in experiments). The model is then tasked with recalling these ten sequences at the end of the context.

**Training and Loss Functions**  We optimize the distillation loss, which is the L2 difference between the last hidden state of the full attention model ($\boldsymbol{H}_{\text{full}}$) and those of the model using DuoAttention ($\boldsymbol{H}_{\text{mixed}}$), focusing only on the last $l$ passkey tokens in the entire inputs with $T$ tokens, where $N$ is the batch size:

$$\mathcal{L}_{\text{distill}} = \frac{1}{N} \sum_{i=1}^{N} \sum_{j=T-l+1}^{T} (\boldsymbol{H}_{\text{full}}^{(i)}[j] - \boldsymbol{H}_{\text{mixed}}^{(i)}[j])^2 \tag{1}$$

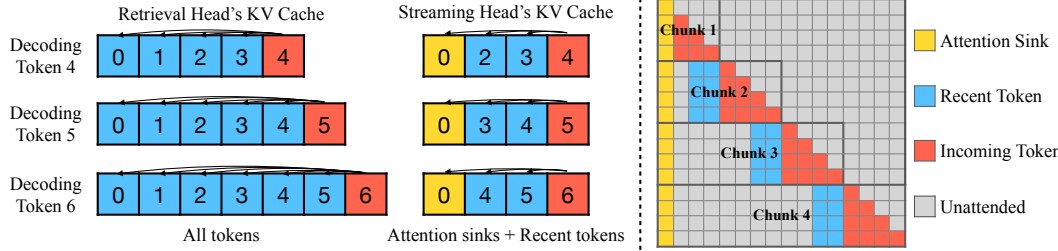

**Figure 5: Decoding (left) and Chunked Pre-filling (right) Processes in DuoAttention:** (1) The retrieval heads' KV cache stores all tokens, while the streaming heads' KV cache retains only recent tokens and attention sinks, ensuring constant memory usage. (2) The chunked pre-filling process of DuoAttention's streaming heads on a 16-token sequence, with one attention sink, two recent tokens, and a chunk size of 4. DuoAttention's streaming heads have linear time and constant memory complexity during long sequence pre-filling.

Our synthetic dataset ensures that every supervision signal is relevant to the final compression strategy, making the process lossless in terms of information retrieval accuracy. It proves to be more effective than using natural language modeling alone (see ablation studies in Section 13). We use the L1 regularization term (a.k.a, Lasso (Tibshirani, 1996)) to encourage sparsity in the gate values:

$$\mathcal{L}_{\text{reg}} = \sum_{i=1}^{\texttt{\#layers}} \sum_{j=1}^{\texttt{\#heads}} |\alpha_{i,j}| . \tag{2}$$

The final training loss is a combination of the distillation loss and the regularization loss, weighted by a hyperparameter $\lambda$, which we set as 0.05 in our experiments:

$$\mathcal{L} = \mathcal{L}_{\text{distill}} + \lambda \mathcal{L}_{\text{reg}} . \tag{3}$$

Since the total number of trainable parameters is only thousands of floating-point numbers, this optimization process is fairly fast, with only 2,000 steps needed. All training experiments in our paper can be conducted on 8×NVIDIA A100 GPU servers.

## 2.3 Deploying LLMs with DuoAttention

**Binarizing Attention Implementations**  At inference time, we apply full attention exclusively to the designated retrieval heads, identified using the optimized gate values from the training phase (as shown in Figure 4). We binarize the attention policy for each head based on a threshold $\tau$, determined by a specified sparsity quantile, to differentiate between retrieval heads and streaming heads:

$$\texttt{attn}_{i,j} = \begin{cases} \texttt{full\_attn} & \text{if } \alpha_{i,j} > \tau \\ \texttt{streaming\_attn} & \text{otherwise} \end{cases} \tag{4}$$

**Reordering Attention Heads**  Before deployment, we preprocess the model by reordering the output channels of the Query, Key, and Value projection weights according to the attention head assignments. This reordering groups retrieval heads and streaming heads into two distinct, consecutive clusters, allowing for efficient slicing and concatenation operations when managing the KV cache for these two types of heads within a layer, rather than relying on scattering and gathering operations.

**Decoding**  As shown in Figure 5, we allocate two KV caches for each layer in the LLM during decoding: one for retrieval heads, which stores all past Keys and Values, and another for streaming heads, which stores only attention sinks and recent tokens, maintaining a constant size. When a new token is processed, its query, key, and value vectors are split along the head dimension to compute full attention for retrieval heads and streaming attention for streaming heads. The results are then concatenated along the head dimension for the output projection.

**Chunked Pre-filling**  We use FlashAttention-2 (Dao, 2023) to pre-fill the KV caches for both retrieval and streaming heads. In long-context LLMs, chunked pre-filling is a common practice (Agrawal et al., 2023; Kwon et al., 2023), dividing the prompt into fixed-length chunks to pre-fill the KV cache. This technique significantly reduces peak memory usage (see Table 10) by lowering the peak intermediate activation size in linear layers from sequence length to chunk size. DuoAttention is fully compatible with chunked pre-filling, and the streaming heads' pre-filling in DuoAttention can be achieved with linear time and constant memory complexity, *without* requiring specialized kernels. As shown in Figure 5, once a layer's KVs are computed, the streaming head's

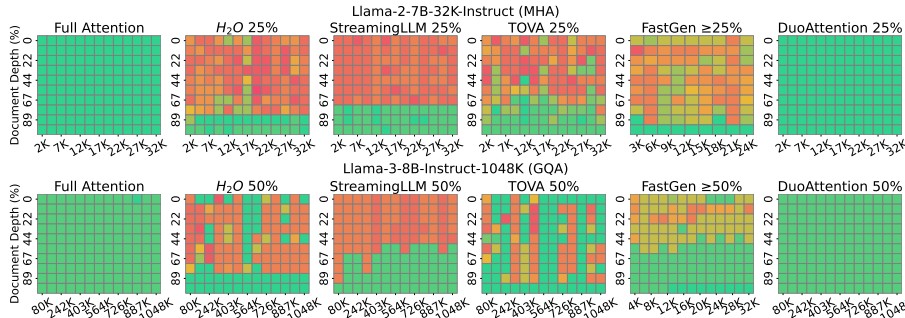

Figure 6: DuoAttention provides comparable accuracy as full attention on the Needle-in-a-Haystack benchmark using 25% full attention ratio on the MHA model and 50% full attention ratio on the GQA model.

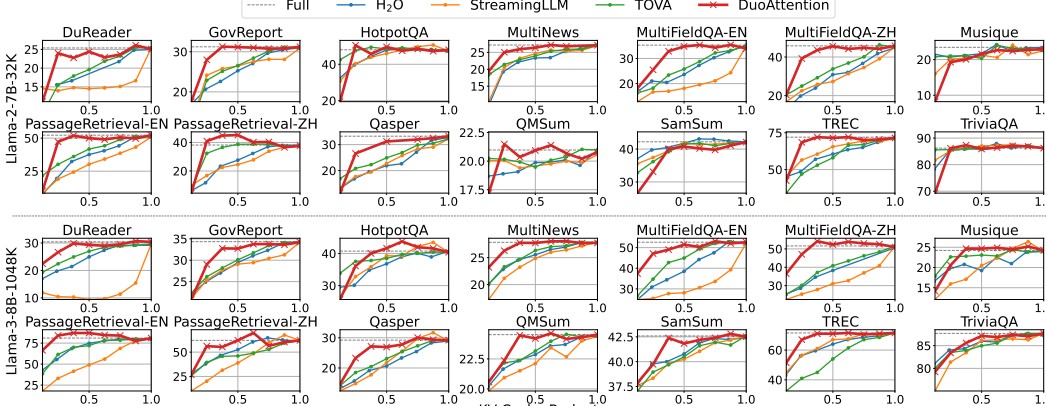

Figure 7: DuoAttention provides better KV budget and accuracy trade-off on LongBench benchmarks.

KV cache is immediately pruned to keep only the sink and recent tokens. The next chunk of incoming tokens will only attend to a constant number of contextual tokens during pre-filling. Let $L$ represent the sequence length and $K$ the chunk size. The pre-filling time complexity for streaming heads is optimized from $O(L^2)$ to $O(LK)$, and the memory complexity is reduced from $O(L)$ to $O(K)$.

It's important to note that DuoAttention's design is well-suited for batch operations, which can further enhance LLM efficiency in serving scenarios with large batch sizes.

## 3 EXPERIMENTS

### 3.1 SETUPS

**Models, Datasets, and Baselines**    We evaluate DuoAttention on both long-context and short-context benchmarks to demonstrate that our method preserves model performance on tasks requiring both long and short contexts while significantly improving efficiency. For long-context evaluations, we use the Needle-in-a-Haystack (NIAH) benchmark (Kamradt, 2024) and LongBench (Bai et al., 2023). For short-context evaluations, we assess performance on MMLU (Hendrycks et al., 2021), MBPP (Austin et al., 2021), and MT-Bench (Zheng et al., 2023). We employ state-of-the-art open-source models, including Llama-2-7B-chat (Touvron et al., 2023b) (and its long-context variant Llama-2-7B-32K-Instruct (Together, 2023)), Llama-3-[8,70]B-Instruct (and its long-context variant Llama-3-8B-Instruct-Gradient-1048k [*]), and Mistral-7B-v0.2-Instruct (Jiang et al., 2023). We compare our method against KV cache compression algorithms, including H2O (Zhang et al., 2023b), TOVA (Oren et al., 2024), FastGen (Ge et al., 2024), and StreamingLLM (Xiao et al., 2023b).

**Implementation Details**    We implement DuoAttention in PyTorch (Paszke et al., 2019) using RoPE (Su et al., 2021) and RMSNorm kernels from FlashInfer (Ye et al., 2024). For retrieval head identification, we use a batch size of 1, inserting ten 32-word passkeys into the BookSum (Kryściński et al., 2021) dataset. The identification process uses 128 sink tokens and 256 recent tokens. Training samples are drawn from 50 intervals ranging from 1,000 tokens to the model-specific maximum length. Passkeys are randomly inserted at 1000 points within the context. Further details are included

---

[*] https://huggingface.co/gradientai/Llama-3-8B-Instruct-Gradient-1048k

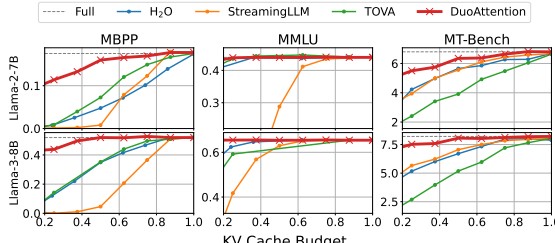

Figure 8: Results on short benchmarks.

Table 1: Llama-3-70B results on short benchmarks.

|  | Budget | MMLU | MBPP | MT-B |
|---|---|---|---|---|
| Full | 100% | 79.38% | 47.85% | 8.93 |
| H2O | 50% | 79.26% | 32.12% | 7.16 |
| TOVA | 50% | 79.15% | 36.09% | 7.96 |
| SLLM | 50% | 77.46% | 5.57% | 5.41 |
| **DuoAttn** | 50% | **79.35%** | **47.09%** | **9.14** |

in Appendix Section A.1. We optimize gate values using the AdamW (Kingma & Ba, 2015) optimizer, starting with a learning rate of 0.02, warming up from 0.002 in the first 400 steps, and reducing back to 0.002 in the final 400 steps. All experiments run for 2,000 steps on NVIDIA A100 GPUs.

## 3.2 LONG-CONTEXT BENCHMARKS

We evaluate DuoAttention using the Needle-in-a-Haystack (NIAH) benchmark and LongBench (Bai et al., 2023). We use two long-context models: Llama-2-7B-32K-Instruct and Llama-3-8B-Instruct-Gradient-1048k. We configure DuoAttention with a 25% retrieval head ratio for Llama-2-7B-32K-Instruct and a 50% ratio for Llama-3-8B-Instruct-Gradient-1048k. We compare DuoAttention with H2O, TOVA, and StreamingLLM using the same KV cache budget. We use 64 sink, 256 recent tokens, and 32,000 pre-filling chunk size for DuoAttention. Since the original designs of H2O and TOVA do not support long contexts, we modify their algorithms by replacing the pre-filling stage with FlashAttention and simulating decoding for the last 50 tokens of the input, following Tang et al. (2024b). FastGen's algorithm does not allow for the specification of the KV compression ratio, as it fluctuates with inputs. Therefore, we adjust the attention recovery ratio to ensure the KV cache budget is, on average, above 25% or 50% in the experiments shown in Figure 6. Additionally, FastGen's quadratic memory cost during the attention profiling phase limits its ability to handle long-context samples. We measure FastGen's performance on NIAH for Llama-2-7B up to a 24K context and for Llama-3-8B up to a 32K context; beyond these sizes, it results in out-of-memory errors. Detailed baseline implementations and justifications are provided in Appendix Section A.3 and Section A.5.

**Needle-in-a-Haystack (NIAH)** is a challenging pressure test designed to assess the ability of models to accurate identify and retrieve relevant information from lengthy context. As shown in Figure 6, all baseline methods fail to retrieve correct answers from the various depths of the long sequence, as they discard the KV cache containing the necessary information during generation. In contrast, DuoAttention retains all KV caches in the retrieval heads while discarding only those in the streaming heads, preserving the model's retrieval capability. As a result, DuoAttention demonstrates strong performance across all sequence depths, handling lengths up to 1048K tokens effectively.

**LongBench** (Bai et al., 2023) is a comprehensive suite of long-context datasets encompassing multiple tasks and natural texts, designed to assess long-context understanding capabilities more thoroughly. Figure 7 shows the performance on 14 LongBench tasks, comparing different methods based on their KV cache budgets. DuoAttention shows a superior trade-off between KV budget and accuracy on most tasks, underscoring its generalizability. Notably, DuoAttention achieves performance comparable to full attention on most tasks, using a 25% KV cache budget for MHA and a 50% KV cache budget for GQA, consistent with the results observed in the needle-in-a-haystack benchmark. We compare DuoAttention with FastGen in Table 5 and 6 in the Appendix. Table 3 and 4 in the Appendix provides full results for all 21 LongBench tasks using the 25% and 50% KV cache budget for the two models, showing that DuoAttention consistently outperforms baselines across most tasks and achieves the highest average scores.

## 3.3 SHORT-CONTEXT BENCHMARKS.

To ensure that DuoAttention does not compromise the model's performance on short-context tasks, we evaluate it alongside all baselines on three short-context benchmarks: MMLU, MBPP, and MT-Bench. These benchmarks assess the model's knowledge, coding abilities, and helpfulness. We use one-shot prompting for MMLU and zero-shot prompting for MBPP and MT-Bench. For DuoAttention, we configure 32 sink tokens and 128 recent tokens on MMLU, and 16 sink tokens and 64 recent tokens on MBPP and MT-Bench. As shown in Figure 8 and Table 1, DuoAttention consistently outperforms all baselines under the same KV cache budget across various models, including Llama-2-7B, Llama-3-8B, and Llama-3-70B-Instruct. With a 50% KV cache budget, DuoAttention achieves near-lossless performance on most benchmarks, demonstrating that it preserves the model's original capabilities.

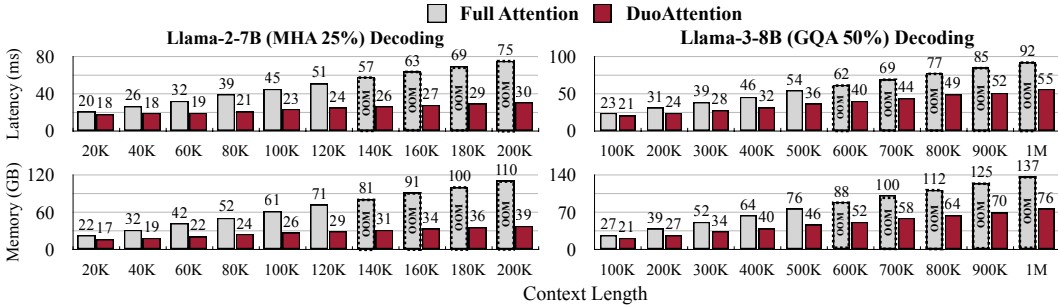

Figure 9: **Per-token decoding latency and memory** usage of DuoAttention compared to full attention across varying **context sizes**. DuoAttention uses a 25% retrieval head ratio for Llama-2-7B (MHA) and 50% for Llama-3-8B (GQA). DuoAttention achieves up to 2.45× memory reduction for MHA and 1.65× for GQA models, along with up to 2.13× latency reduction for MHA and 1.5× for GQA models. These reductions approach the inverse of the retrieval head ratios as context length increases. Out-of-memory (OOM) results are linearly extrapolated from measured data.

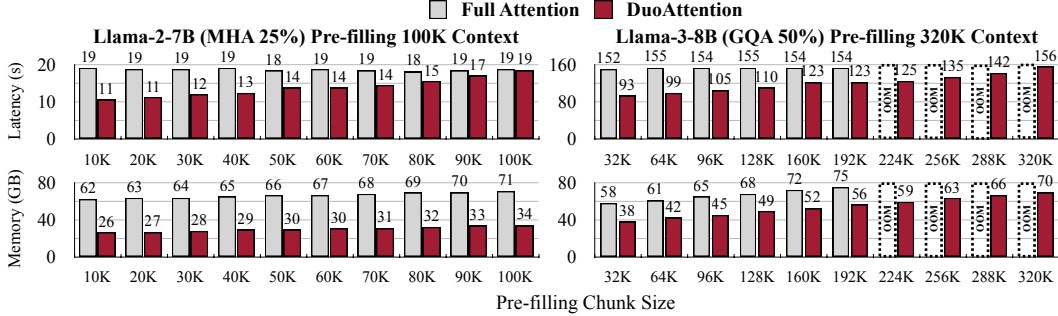

Figure 10: **Pre-filling latency and memory** usage of DuoAttention compared to full attention across varying **pre-filling chunk sizes**. DuoAttention uses a 25% retrieval head ratio for Llama-2-7B (MHA), pre-filling a context of 100K tokens, and a 50% ratio for Llama-3-8B (GQA), pre-filling a context of 320K tokens. As the pre-filling chunk size decreases, DuoAttention achieves up to 1.73× latency reduction for MHA and 1.63× for GQA models, with memory reductions up to 2.38× for MHA and 1.53× for GQA models.

## 3.4 EFFICIENCY RESULTS

We evaluate DuoAttention's decoding latency and memory usage on Llama-2-7B and Llama-3-8B models on a single NVIDIA A100 GPU. We pre-allocate the KV cache for the entire benchmark sequence to prevent the extra overheads of dynamic memory allocations. The default number format for weights and activations is BFloat16. By employing a retrieval head ratio of 25% for Llama-2-7B and 50% for Llama-3-8B, DuoAttention maintains accuracy while significantly improving efficiency.

**Decoding Efficiency**   As shown in Figure 9, DuoAttention's decoding speed scales linearly, though with a flatter slope compared to full attention, reflecting the chosen retrieval head ratio. This efficient scaling leads to significant reductions in memory usage and notable improvements in decoding speed. These improvements approach the inverse of the retrieval head ratios as context length increases. Figure 11 shows DuoAttention's speedup and memory savings across various KV budget settings for a fixed context size. Both decoding latency and memory usage decrease linearly as the ratio of retrieval heads is reduced in the deployment configuration. Under the settings in Figure 11, DuoAttention achieves maximum improvements on an A100 GPU: 2.55× memory reduction for MHA and 1.67× for GQA models, and 2.18× latency reduction for MHA and 1.50× for GQA models.

**Pre-filling Efficiency**   DuoAttention also accelerates long-context pre-filling for LLMs, as discussed in Section 2.3. Figure 10 shows that DuoAttention significantly reduces both pre-filling latency and memory usage, with these savings increasing as the pre-filling chunk size decreases. This is because the time and memory complexity for the streaming heads are reduced with smaller chunk sizes. DuoAttention achieves up to 1.73× latency reduction for MHA and 1.63× for GQA models, with memory reductions of up to 2.38× for MHA and 1.53× for GQA models.

**Combiniation with Quantization**   To fit more tokens into limited memory, we can integrate weight and KV cache quantization with DuoAttention to maximize KV cache capacity. Previous studies

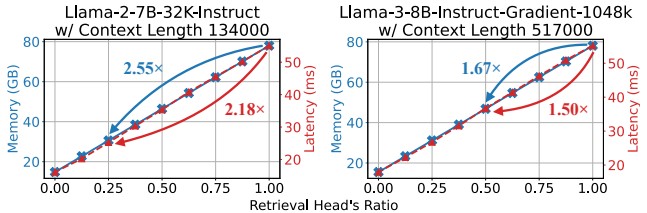 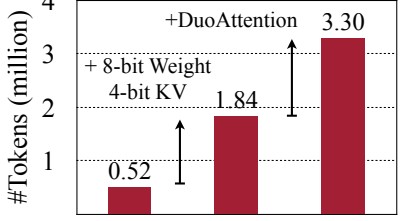

Figure 11: DuoAttention's decoding memory and latency *vs.* KV budget with a fixed context length. Memory and latency are reduced linearly when the ratio of retrieval heads is reduced. DuoAttention achieves up to 2.55× memory reduction for MHA and 1.67× for GQA models, along with up to 2.18× latency reduction for MHA and 1.50× for GQA models.

Figure 12: Combined with 8-bit weight and 4-bit KV cache quantization, DuoAttention can accommodate 3.30 million tokens on a single A100-80G GPU for the Llama-3-8B model.

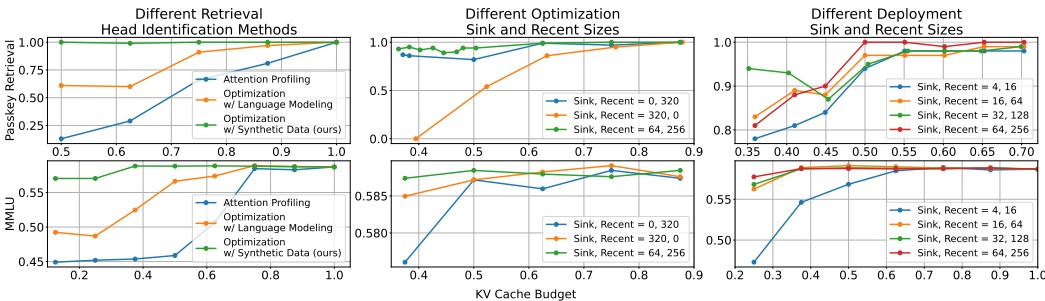

Figure 13: Ablation studies: (1) Comparison of retrieval head identification methods, showing the superiority of our optimization-based approach with synthetic data over attention profiling and language modeling. (2) Analysis of start and recent token sizes shows that combining sink and recent attention optimally identifies retrieval heads. (3) Deployment performance indicates 16 attention sinks and 64 recent tokens are optimal, with minimal gains beyond these values.

have shown that weight quantization (Xiao et al., 2023a; Lin et al., 2024) and 4-bit KV cache quantization (Lin* et al., 2024; Liu et al., 2024; Hooper et al., 2024) do not compromise model performance. We combine DuoAttention with the QServe (Lin* et al., 2024) quantization method and kernels to enable 8-bit weight and 4-bit KV cache LLM inference. Measured results are shown in Figure 12. Combining quantization techniques with DuoAttention allows us to accommodate up to 3.30 million tokens on a single A100-80G GPU using the Llama-3-8B model, resulting in a 6.4× increase in capacity compared to the naive full attention BF16 deployment.

## 3.5 ABLATION STUDIES

We conduct ablation studies using the Mistral-7B-Instruct-v0.2 on passkey retrieval and MMLU datasets. For the passkey retrieval task, we embed an 8-word passkey within a 30K-word text and perform a linear sweep across 100 insertion depths, reporting exact match accuracies.

**Optimization-based vs. Attention Profiling-based Retrieval Head Identification**  We assess our optimization-based method against attention profiling, as used in FastGen (Ge et al., 2024) and RazorAttention (Tang et al., 2024a), utilizing the same synthetic passkey dataset for both. Results in Figure 13 (1) show our method significantly outperforms attention profiling, which struggles to identify retrieval heads, affecting model optimization accurately.

**Optimizing with Synthetic Data vs. Language Modeling**  As illustrated in Figure 13 (1), our approach of using synthetic data to identify retrieval heads produces significantly better results than traditional language modeling, which computes loss on all tokens in natural data.

**Necessity of Sink+Recent Attention in Optimization**  Figure 13 (2) highlights the importance of combining sink and recent attention during the optimization phase. Exclusive reliance on either starting or recent token attention is inadequate for effective retrieval head identification.

**Deployment Phase Configuration**  We analyze the deployment configuration for attention sinks and recent tokens within streaming heads. Our findings indicate that performance plateaus at 16 sink tokens and 64 recent tokens (Figure 13 (3)). Further increases yield marginal improvements.

## 4 RELATED WORK

Various approaches have been developed to scale up LLMs and improve their efficiency in handling long contexts. These methods can be grouped into four main categories: optimizing model architectures, using approximate attention mechanisms, applying KV cache quantization, and system-level optimizations.

**Model Architecture**   Multi-Query Attention (MQA)(Shazeer, 2019) and Grouped-Query Attention (GQA)(Ainslie et al., 2023) reduce the size of the Key-Value (KV) cache by sharing KV heads across query heads. However, these methods require pre-training with specific architectures and do not reduce computational costs. Linear attention Transformers (Gu & Dao, 2023) reduce memory usage but tend to underperform on tasks requiring long-context processing.

**Approximate Attention**   Methods like Sparse Transformer (Child et al., 2019) and Long-Former (Beltagy et al., 2020) use local or block attention patterns to reduce computational complexity. BigBird (Zaheer et al., 2020) achieves linear complexity by combining local and global attention, but many of these methods require custom GPU kernels or retraining, limiting their practicality. H2O (Zhang et al., 2023b) and TOVA (Oren et al., 2024) simplify attention by discarding tokens based on query patterns. StreamingLLM (Xiao et al., 2023b) identifies "attention sinks" and proposes always retaining initial and recent tokens to maintain constant decoding latency and memory usage, allowing the model to process significantly more input tokens than the pre-training sequence length. FastGen (Ge et al., 2024) profiles attention heads to discard tokens during decoding. However, our experiments show that these methods degrade the long-context abilities of LLMs. Also, methods like H2O and TOVA cannot reduce the pre-filling cost of long-context LLMs.

**KV Cache Quantization**   Techniques such as 8-bit and 4-bit quantization (Liu et al., 2024; Hooper et al., 2024; Lin* et al., 2024) reduce the size of KV caches, but they do not address the computational overhead of attention kernels. These methods are complementary to DuoAttention and can be used together to further reduce memory usage.

**System Optimizations**   vLLM (Kwon et al., 2023) and FlashAttention (Dao et al., 2022; Dao, 2023) improve attention computation efficiency by optimizing batch processing and utilizing GPU memory hierarchies. FlashDecoding (Hong et al., 2024) and RingAttention (Liu et al., 2023a) introduce further improvements in decoding speed and sequence-level parallelism. While these methods enhance computational performance, they do not address KV cache size reduction, making them complementary to DuoAttention for additional speed and memory optimization.

**Recent Works**   Several recent works share similar ideas with DuoAttention. Wu et al. (2024) introduces the concept of retrieval heads to explain LLMs' long-context capabilities. However, their approach does not compress the KV cache for non-retrieval heads, focusing solely on accuracy. MInference (Jiang et al., 2024) accelerates pre-filling for long-context LLMs by using sparse attention patterns but does not optimize KV cache storage or latency during decoding. RazorAttention (Tang et al., 2024a) also divides attention heads into retrieval and non-retrieval categories but relies on attention profiling, which, as our experiments show, is less accurate than our optimization-based approach. Also, RazorAttention doesn't optimize pre-filling. DuoAttention offers more effective KV cache management and higher compression rates, leading to better performance for both pre-filling and decoding in long-context applications.

## 5 CONCLUSION

We introduce DuoAttention, a framework that optimizes memory and computational resources in LLMs by distinguishing between *Retrieval Heads* and *Streaming Heads*. By applying a full KV cache only to retrieval heads, DuoAttention significantly reduces memory usage and latency for both decoding and pre-filling in long-context applications. It achieves memory reductions of up to $2.55\times$ for MHA and $1.67\times$ for GQA models, with decoding speed improvements of up to $2.18\times$ for MHA and $1.50\times$ for GQA, and pre-filling accelerations of up to $1.73\times$ and $1.63\times$, respectively, with minimal accuracy loss compared to full attention. When combined with quantization, DuoAttention further boosts KV cache capacity, supporting up to 3.30 million contextual tokens on a single A100 GPU. DuoAttention paves the way for LLMs to handle contexts with millions of tokens.

ACKNOWLEDGMENTS

We thank MIT-IBM Watson AI Lab, MIT and Amazon Science Hub, MIT AI Hardware Program, National Science Foundation, Hyundai, and Samsung for supporting this research. We thank NVIDIA for donating the DGX server.

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

# A    APPENDIX

## A.1    EXPERIMENTAL DETAILS

We use FSDP2 in PyTorch for model training and DeepSpeed Ulysses (Jacobs et al., 2023) sequence parallelism to support long sequences. During training, we use an efficient block-sparse approximation of $\Lambda$-like attention for streaming attention, as implemented in Guo et al. (2024) and illustrated in Figure 14. Maximum sequence lengths vary across models, as detailed in Table 2.

Table 2: Training Hyperparameters.

| Models | Max. Seq. Lengths |
| --- | --- |
| Llama-2-7B-chat | 4096 |
| Llama-2-7B-32K-Instruct | 32000 |
| Llama-3-8B-Instruct | 8192 |
| Llama-3-8B-Instruct-1048K | 32000 |
| Llama-3-70B-Instruct | 8192 |
| Mistral-7B-Instruct-v0.2 | 32000 |

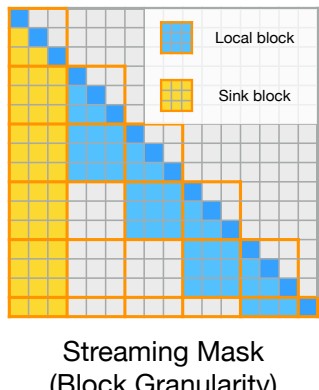

Streaming Mask
(Block Granularity)

Figure 14: Block-sparse approximation of $\Lambda$-like attention.

## A.2    FULL LONGBENCH RESULTS

Table 3 and Table 4 show the full LongBench results of DuoAttention and baselines.

## A.3    IMPLEMENTATION OF H2O AND TOVA ON LONG-CONTEXT BENCHMARKS

The original designs of the H2O and TOVA algorithms are not compatible with FlashAttention during pre-filling, as they rely on attention scores to perform token eviction. Since attention scores in FlashAttention are never materialized, these algorithms cannot be used in pre-filling, which is one of their main flaws. Therefore, it's not possible to evaluate these algorithms in long-context settings like needle-in-the-haystack and LongBench, as they cause OOM during context pre-filling. To compare with these strategies, we modified the algorithms: during pre-filling, we used FlashAttention for exact calculations. During the decoding stage, we perform token eviction based on the generated tokens' attention scores to contextual tokens. This modification improves performance compared to the original design since pre-filling is exact and token eviction occurs only during decoding. In extreme scenarios, if there is only one generated token in the answer (e.g. multiple-choice tasks), our implementation of H2O and TOVA will be exact with full attention, unlike their true accuracy. To approach their true performance, we simulate the last 50 tokens in long input benchmarks (needle-in-the-haystack and LongBench) as generated tokens to perform their token eviction policy long enough, as well as our algorithm. This experimental setting is also used by Tang et al. (2024b). Experimental results show our method can pass this pressure test, while H2O and TOVA cannot.

Table 3: Full LongBench results with Llama-3-8B-Instruct-1048K. DuoAttention achieves the best performance with a 50% KV cache budget on most datasets.

| Dataset | Full | H2O (50%) | SLLM (50%) | TOVA (50%) | **Duo (50%)** |
|---|---|---|---|---|---|
| **Average** | 40.08 | 35.76 | 32.26 | 35.55 | **40.21** |
| 2WikiMQA | 28.78 | 27.99 | **29.22** | 26.93 | 29.08 |
| DuReader (zh) | 30.41 | 24.94 | 9.41 | 27.00 | **29.31** |
| GovReport | 34.23 | 29.44 | 29.08 | 30.10 | **32.72** |
| HotpotQA | 40.37 | 36.77 | 39.27 | 38.45 | **41.63** |
| LCC | 38.19 | 43.09 | 41.94 | 42.31 | **44.16** |
| LSHT (zh) | 38.00 | 25.00 | 25.50 | 24.50 | **30.00** |
| MultiNews | 27.73 | 25.52 | 24.85 | 26.32 | **27.72** |
| MultiFieldQA-en | 52.62 | 38.53 | 28.11 | 44.94 | **51.44** |
| MultiFieldQA-zh | 50.58 | 38.25 | 31.07 | 40.82 | **52.40** |
| Musique | 24.22 | 19.24 | 20.47 | 23.07 | **24.65** |
| NarrativeQA | 26.56 | 25.13 | 22.06 | **25.64** | 24.54 |
| Passage Count | 1.00 | **2.05** | 1.64 | 1.00 | 0.00 |
| PassageRetrieval-en | 81.00 | 74.75 | 49.00 | 72.00 | **87.00** |
| PassageRetrieval-zh | 62.15 | 52.57 | 38.90 | 46.13 | **62.15** |
| Qasper | 29.21 | 20.65 | 21.77 | 23.06 | **26.93** |
| QMSum | 24.52 | 22.87 | 22.11 | 23.16 | **24.20** |
| RepoBench-P | 38.94 | 39.98 | 37.60 | 40.14 | **46.12** |
| SAMSum | 42.51 | 40.78 | 40.25 | 40.50 | **41.83** |
| TREC | 71.50 | 64.00 | 67.00 | 54.00 | **71.00** |
| TriviaQA | 87.70 | 85.98 | 86.11 | 84.97 | **87.14** |
| VCSUM (zh) | 11.37 | **13.45** | 12.10 | 11.59 | 10.46 |

## A.4 NIAH RESULTS ON MISTRAL MODELS

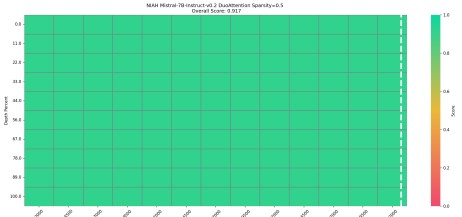
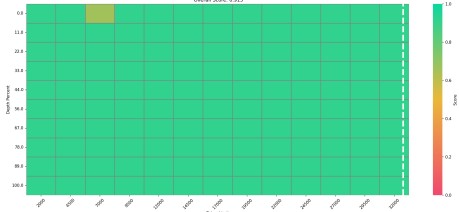

Figure 15: NIAH result on the Mistral-7B-Instruct-v0.2 model.

Figure 16: NIAH result on the Mistral-7B-Instruct-v0.3 model.

## A.5 IMPLEMENTATION OF FASTGEN ON LONG-CONTEXT BENCHMARKS

Due to the lack of official implementation of the FastGen (Ge et al. (2024)) algorithm, we reproduce it using a community codebase (Adams et al. (2024)), which is referenced by FastGen's official repository. In the FastGen algorithm, the pruning ratio cannot be directly configurable; instead, the recovery ratio $T$ is used to control sparsity as outlined in the FastGen paper. To quantify sparsity, we calculated the average KV cache usage across all test cases as the overall measure of sparsity. For the Llama-2-7B model, we set the recovery ratio to $0.7$, ensuring the average KV cache budget was over 25% of the full KV cache. Similarly, for the Llama-3-8B model, we set the recovery ratio to $0.87$, ensuring the average KV cache budget was more than 50% of the full KV cache. Additionally, since FastGen uses the full attention map of the user-provided prompt to profile the types of different heads, it results in an $O(n^2)$ attention map complexity. Therefore, we are unable to test its performance in long contexts. For the long context benchmark, we used 8 A100-80G GPUs, achieving sequence lengths of up to 24k tokens for the Llama-2-7B model and up to 32k tokens for the Llama-3-8B model. In addition to the needle-in-the-haystack benchmark shown in Figure 6, we also evaluated

Table 4: Full LongBench results with Llama-2-7B-Instruct-32K. DuoAttention achieves the best performance with a 25% KV cache budget on most datasets.

| Dataset | Full | H2O (25%) | SLLM (25%) | TOVA (25%) | **Duo (25%)** |
|---|---|---|---|---|---|
| **Average** | 37.52 | 26.84 | 27.80 | 29.78 | **34.49** |
| 2WikiMQA | 35.59 | 28.87 | 29.69 | 31.18 | **33.37** |
| DuReader (zh) | 25.10 | 15.56 | 13.96 | 15.51 | **23.99** |
| GovReport | 31.23 | 20.66 | 24.14 | 22.88 | **27.98** |
| HotpotQA | 47.98 | 39.60 | 40.39 | 47.45 | **50.44** |
| LCC | 51.21 | 45.78 | 44.25 | 47.91 | **48.34** |
| LSHT (zh) | 34.50 | 16.50 | 17.50 | 18.50 | **25.50** |
| MultiNews | 27.11 | 19.21 | 20.54 | 21.41 | **25.03** |
| MultiFieldQA-en | 33.95 | 21.01 | 16.69 | 18.19 | **25.49** |
| MultiFieldQA-zh | 45.79 | 19.81 | 22.50 | 24.96 | **39.23** |
| Musique | 22.97 | 20.63 | 20.09 | **21.00** | 19.27 |
| NarrativeQA | 24.11 | 19.14 | 21.13 | **23.06** | 20.49 |
| Passage Count | 0.00 | 0.53 | **0.58** | 0.00 | 0.33 |
| PassageRetrieval-en | 50.92 | 19.50 | 19.08 | 30.17 | **47.25** |
| PassageRetrieval-zh | 37.68 | 11.75 | 16.77 | 32.38 | **40.93** |
| Qasper | 33.23 | 16.84 | 17.68 | 20.85 | **26.59** |
| QMSum | 20.79 | 18.89 | 20.05 | 20.16 | **21.48** |
| RepoBench-P | 51.58 | 45.16 | 45.25 | **49.03** | 48.58 |
| SAMSum | 42.10 | **39.73** | 37.43 | 36.17 | 33.10 |
| TREC | 71.50 | 48.50 | 56.50 | 47.00 | **68.50** |
| TriviaQA | 86.21 | 85.16 | 85.24 | 85.65 | **86.15** |
| VCSUM (zh) | 14.45 | 10.71 | **14.36** | 11.85 | 12.35 |

Table 5: Comparison of FastGen and DuoAttention on a subset of LongBench using the Llama-3-8B-Instruct-1048K model.

| | FastGen (>50%) | **DuoAttention (50%)** |
|---|---|---|
| Average | 32.82 | **40.01** |
| 2WikiMQA | 18.61 | **29.08** |
| DuReader (zh) | 20.22 | **29.31** |
| HotpotQA | 33.08 | **41.63** |
| LCC | **46.50** | 44.16 |
| MultiNews | 18.18 | **27.72** |
| MultiFieldQA-en | 44.05 | **51.44** |
| MultiFieldQA-zh | 42.15 | **52.40** |
| Musique | 13.58 | **24.65** |
| Passage Count | 0.09 | 0.00 |
| PassageRetrieval-en | **93.12** | 87.00 |
| PassageRetrieval-zh | 40.75 | **62.15** |
| Qasper | 26.51 | **26.93** |
| QMSum | 24.03 | **24.20** |
| SAMSum | 34.12 | **41.83** |
| TriviaQA | 69.92 | **87.14** |
| VCSUM (zh) | 0.23 | **10.46** |

FastGen on LongBench for both models. However, due to the quadratic memory consumption of FastGen, we only report results for datasets that were feasible to run on 8x A100-80G GPUs using FastGen. As shown in Table 5 and Table 6, DuoAttention can consistently outperform FastGen on LongBench datasets.

Table 6: Comparison of FastGen and DuoAttention on a subset of LongBench using the Llama-2-7B-32K-Instruct model.

|  | FastGen (>25%) | **DuoAttention (25%)** |
|---|---|---|
| Average | 19.01 | **32.81** |
| 2WikiMQA | 28.05 | **33.37** |
| MultiNews | 12.60 | **25.03** |
| MultiFieldQA-en | **28.58** | 25.49 |
| MultiFieldQA-zh | 22.44 | **39.23** |
| PassageRetrieval-zh | 3.38 | **40.93** |

## A.6    COMPARISON WITH RECENT KV CACHE COMPRESSION METHODS (SNAPKV, PYRAMIDKV)

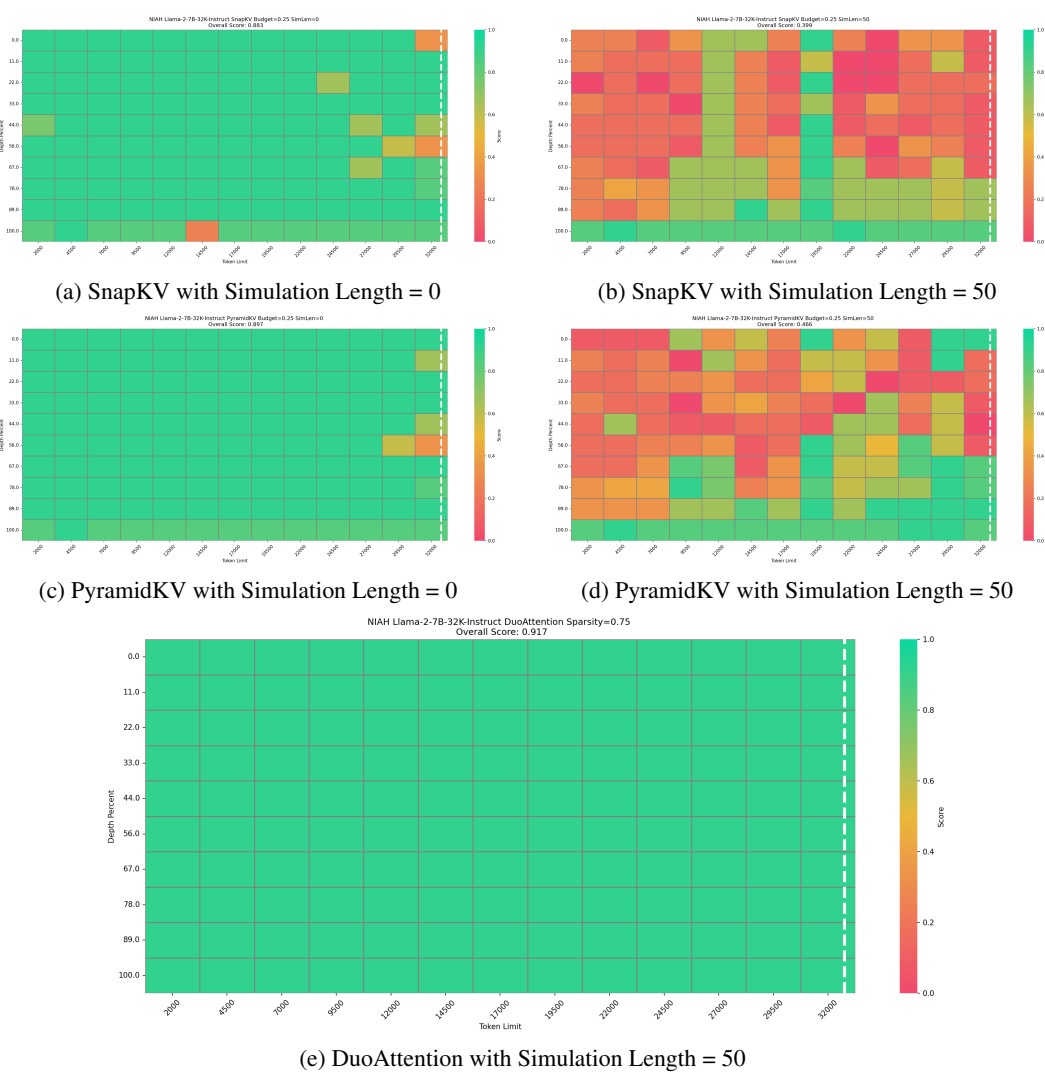

(a) SnapKV with Simulation Length = 0

(b) SnapKV with Simulation Length = 50

(c) PyramidKV with Simulation Length = 0

(d) PyramidKV with Simulation Length = 50

(e) DuoAttention with Simulation Length = 50

Figure 17: NIAH results for Llama-2-7B-32K-Instruct with a 25% KV cache budget.

SnapKV (Li et al., 2024) and PyramidKV (Cai et al., 2024) are recent KV cache compression methods that use a local window of observed tokens to determine which KV cache tokens to retain. Both methods rely on computing attention scores for the last few tokens (typically 8–64) over the entire

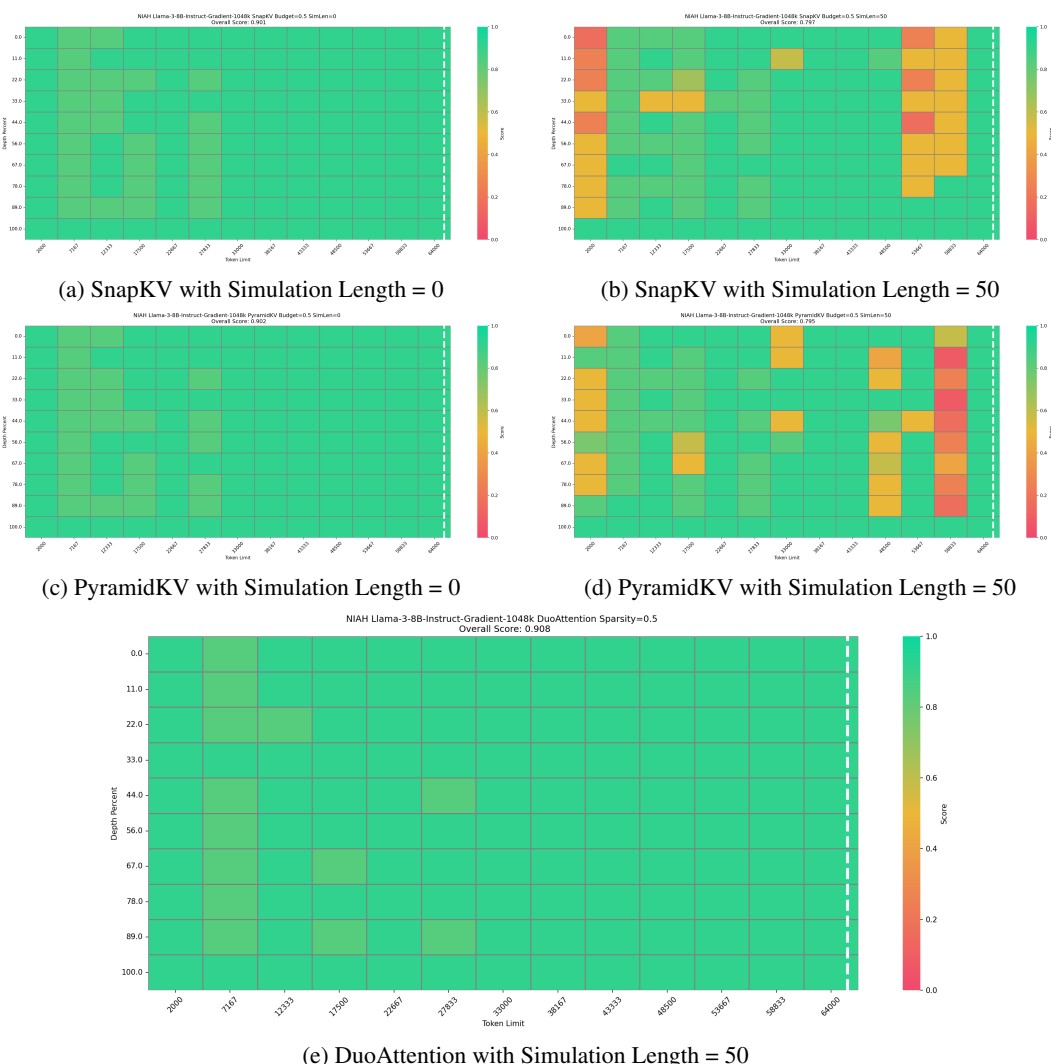

(a) SnapKV with Simulation Length = 0      (b) SnapKV with Simulation Length = 50

(c) PyramidKV with Simulation Length = 0      (d) PyramidKV with Simulation Length = 50

(e) DuoAttention with Simulation Length = 50

Figure 18: NIAH results for Llama-3-8B-Instruct-Gradient-1048k with a 50% KV cache budget.

context and pruning tokens based on these scores. This approach performs well on benchmarks like Needle-in-a-Haystack (NIAH) and LongBench, where queries appear at the end of the prompt.

However, these methods assume that critical query information is located at the end of the context, which is not always valid in real-world scenarios such as multi-turn dialogues or tasks where queries are positioned earlier in the prompt. This reliance reduces their flexibility and general applicability.

Figures 17 and 18 compare the performance of SnapKV and PyramidKV with DuoAttention under equivalent KV cache budget constraints (25% for Llama-2-7B-32K-Instruct and 50% for Llama-3-8B-Instruct-Gradient-1048k). The evaluations include both cases: without simulating the last tokens as generated tokens (Simulation Length = 0) and with simulation of the last 50 tokens as generated inputs (Simulation Length = 50, mimicking a second-round dialogue scenario). Details of the testing procedure are provided in Appendix Section A.3.

As shown, DuoAttention performs comparably or better than SnapKV and PyramidKV when no simulation is applied. However, when the last 50 tokens are treated as generated inputs, SnapKV and PyramidKV experience severe accuracy drops, even under large KV cache budgets. This failure occurs because these methods rely on observing the final tokens to guide pruning, which breaks under these conditions. In contrast, DuoAttention maintains robust accuracy under the same stress test.

These results highlight DuoAttention as a more general and robust KV cache compression method, capable of adapting to diverse real-world scenarios without relying on assumptions about token positions within the context.

## A.7 COMBINATION WITH PRE-FILLING ACCELERATION METHODS (MINFERENCE)

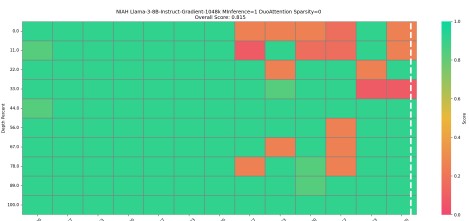
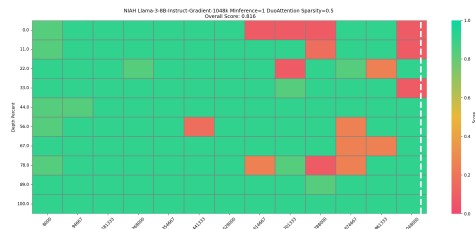

Figure 19: MInference applied to all attention heads.

Figure 20: DuoAttention + MInference applied to retrieval heads.

MInference (Jiang et al., 2024) employs sparsity patterns, such as block-sparse and vertical-slash patterns, observed within token windows to *accelerate pre-filling*. However, it is limited to the pre-filling stage and does not improve decoding speed or reduce the KV cache size.

We demonstrate that MInference is an orthogonal method that can complement DuoAttention by further accelerating the pre-filling stage of retrieval heads. As shown in Figures 19 and 20, applying MInference alone on our NIAH benchmark results in some accuracy degradation compared to full attention or pure DuoAttention (refer to Figure 6).

By combining MInference with DuoAttention, we replace half of the attention heads in LLMs with streaming heads. This approach maintains comparable accuracy while achieving significant reductions in both the KV cache size (nearly halved) and decoding overhead. These results highlight the compatibility and efficiency of combining DuoAttention with MInference.

## A.8 RESULTS ON RULER

RULER (Hsieh et al., 2024) is a synthetic dataset designed to rigorously evaluate long-context language models with configurable sequence lengths and task complexities. It includes 13 tasks spanning 4 categories, assessing long-context capabilities beyond simple in-context recall.

Table 7 presents the average accuracy of full attention and DuoAttention (50% sparsity) across different context lengths, using the Llama-3-8B-Instruct-Gradient-1048k model for sequences up to 128K. The results demonstrate that DuoAttention achieves accuracy scores comparable to full attention across all context lengths, with even an average performance increase of 0.05%.

Table 7: RULER results comparing full attention and DuoAttention using the Llama-3-8B-Instruct-Gradient-1048k model.

| Context Length | 4K | 8K | 16K | 32K | 64K | 128K | Avg. |
|---|---|---|---|---|---|---|---|
| **Full Attention** | 92.78 | 90.54 | 86.41 | 80.59 | 76.33 | 73.01 | 83.28 |
| **DuoAttention (50%)** | 92.83 | 91.17 | 85.17 | 81.28 | 75.81 | 73.71 | 83.33 |

These findings validate DuoAttention 's effectiveness in maintaining strong accuracy on a rigorous benchmark, even under more challenging long-context evaluation settings.

## A.9    ACCURACY RESULTS WHEN COMBINING WITH QUANTIZATION

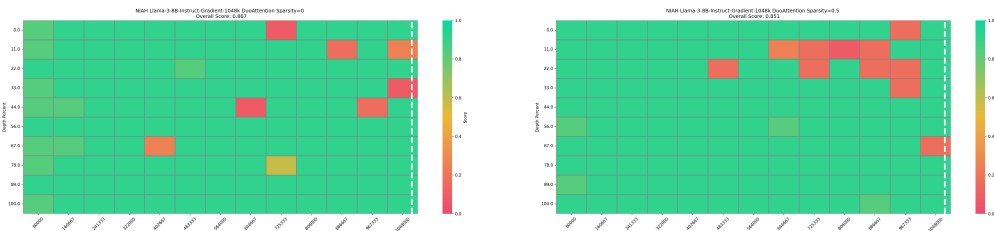

Figure 21: Full Attention with INT4 KV Cache     Figure 22: DuoAttention with INT4 KV Cache

We conducted experiments to evaluate the performance of combining DuoAttention with KV quantization. Specifically, we examined two configurations:

1. **Baseline:** The original model with INT4 KV Pre-Rope quantization and a group size of 128, as proposed in KIVI (Liu et al., 2024) (see Figure 21).

2. **Proposed Combination:** The model incorporating DuoAttention with 50% sparsity alongside the same INT4 KV Pre-Rope quantization (see Figure 22).

For this study, we utilized the Llama-3-8B-Instruct-Gradient-1048k model. Notably, both the full attention model and the DuoAttention-enabled model achieve perfect accuracy when using FP16 KV caches (refer to Figure 6).

The key results are as follows:

- **Baseline (INT4 KV Pre-Rope Quantization):** The model achieves an overall accuracy score of 0.867, demonstrating a slight accuracy drop compared with using the FP16 KV cache (Figure 21).

- **DuoAttention + INT4 KV Quantization:** The combined approach achieves an overall accuracy score of 0.851, reflecting only a minor reduction of 0.016 in performance relative to the INT4 KV baseline (Figure 22).

These findings confirm that incorporating DuoAttention (with 50% sparsity) has a negligible impact on overall accuracy while offering potential computational advantages. This validates the efficacy of the combined approach in preserving accuracy while optimizing resource efficiency.

## A.10    RESULTS ON THE LLAMA-3.1 MODEL

Table 8 shows the LongBench results on the Llama-3.1-8B-Instruct model. The trends are consistent with Llama-3-8B-Instruct-Gradient-1048k used in the main text, with DuoAttention achieving accuracy comparable to full attention and outperforming baselines.

Table 8: LongBench results with Llama-3-8.1B-Instruct. DuoAttention achieves accuracy comparable to full attention and outperforms baselines.

| Dataset | Full | H2O (50%) | SLLM (50%) | TOVA (50%) | **Duo (50%)** |
|---|---|---|---|---|---|
| **Average** | 39.01 | 35.61 | 31.32 | 36.18 | **38.91** |
| 2WikiMQA | 16.37 | 13.91 | 13.25 | 14.22 | **16.20** |
| DuReader (zh) | 29.30 | 21.53 | 12.95 | 22.07 | **31.31** |
| GovReport | 34.53 | 30.56 | 30.47 | 30.78 | **32.87** |
| HotpotQA | 17.23 | 17.31 | 15.78 | 16.29 | **19.53** |
| LCC | 52.39 | 53.08 | 52.90 | 52.39 | **53.31** |
| LSHT (zh) | 46.00 | 39.00 | 36.00 | 42.50 | **45.00** |
| MultiNews | 26.91 | 25.52 | 24.97 | 25.14 | **26.29** |
| MultiFieldQA-en | 28.44 | 21.89 | 16.05 | 21.59 | **27.77** |
| MultiFieldQA-zh | 20.19 | 14.87 | 15.92 | 16.55 | **21.98** |
| Musique | 11.82 | 10.15 | 10.19 | 9.64 | **12.97** |
| NarrativeQA | 31.99 | 31.09 | 24.15 | **31.56** | 29.12 |
| Passage Count | 6.26 | 5.40 | 4.75 | **6.68** | 6.31 |
| PassageRetrieval-en | 97.95 | 89.86 | 52.11 | 97.44 | **98.59** |
| PassageRetrieval-zh | 77.54 | 69.73 | 35.14 | 71.81 | **75.37** |
| Qasper | 25.14 | 16.96 | 23.56 | 20.75 | **21.12** |
| QMSum | 23.63 | 22.54 | 21.48 | 22.82 | **23.89** |
| RepoBench-P | 49.46 | 49.51 | 49.95 | 49.36 | **53.74** |
| SAMSum | 43.69 | 42.56 | 43.32 | 42.28 | **43.40** |
| TREC | 72.50 | 66.50 | 69.50 | 58.00 | **73.00** |
| TriviaQA | 91.65 | 90.07 | 90.06 | **91.73** | 89.60 |
| VCSUM (zh) | 16.26 | 15.80 | 15.17 | 16.09 | **15.83** |

## A.11 LONGBENCH RESULTS COMPARING WITH SNAPKV, PYRAMIDKV, AND ADAKV

Table 9 presents a detailed comparison of DuoAttention, SnapKV Li et al. (2024), PyramidKV Cai et al. (2024), and AdaKV Feng et al. (2024) under a consistent 50% KV cache budget using the Llama-3-8B-Instruct-Gradient-1048k model, on LongBench.

DuoAttention achieves the highest average performance (40.21), outperforming SnapKV, PyramidKV, and AdaKV on most datasets. While SnapKV, PyramidKV, and AdaKV rely on an observation window to determine relevant KV cache entries, making them sensitive to query positioning, DuoAttention does not depend on this heuristic. This allows it to perform robustly across various scenarios, including continuous pre-filling and multi-round dialogue, where queries are not always positioned at the end of the context. Furthermore, DuoAttention demonstrates strong generalization across different tasks, maintaining higher accuracy under constrained KV cache budgets. These results highlight its applicability to real-world retrieval and reasoning tasks.

## A.12 IMPLEMENTATION DETAILS OF THE NEEDLE-IN-THE-HAYSTACK BENCHMARK

Our implementation follows the setup of the original Needle-in-the-Haystack benchmark Kamradt (2024). The haystack corpus is constructed by concatenating Paul Graham's essays. The "needle" inserted into this haystack is the text:

"Remember, the best thing to do in San Francisco is eat a sandwich and sit in Dolores Park on a sunny day."

The corresponding retrieval question is:

  "What is the best thing to do in San Francisco?Answer: The best thing to do in San Francisco is"

For evaluation, we calculate a score based on the word-level overlap between the model's response and the expected output. Specifically, let `model_response` denote the model's response and `expected_answer` represent the target output split into individual words, which is:

"eat a sandwich and sit in Dolores Park on a sunny day."

Table 9: Comparison of DuoAttention, SnapKV, PyramidKV, and AdaKV under a 50% KV cache budget using the Llama-3-8B-Instruct-Gradient-1048k model. DuoAttention achieves the highest accuracy across tasks.

| Dataset | Full | SnapKV | PyramidKV | AdaKV | **DuoAttention** |
|---|---|---|---|---|---|
| **Average** | 40.08 | 38.47 | 38.39 | 38.67 | **40.21** |
| 2WikiMQA | 28.78 | 29.00 | 28.12 | 28.97 | **29.08** |
| DuReader (zh) | 30.41 | 24.04 | 26.63 | 22.65 | **29.31** |
| GovReport | 34.23 | 26.84 | 27.59 | 24.22 | **32.72** |
| HotpotQA | 40.37 | 40.86 | 41.56 | 40.23 | **41.63** |
| LCC | 38.19 | 38.83 | 37.59 | 39.67 | **44.16** |
| LSHT (zh) | 38.00 | 38.00 | **38.50** | 36.50 | 30.00 |
| MultiNews | 27.73 | 22.84 | 22.93 | 21.81 | **27.72** |
| MultiFieldQA-en | 52.62 | 51.96 | 52.54 | **52.99** | 51.44 |
| MultiFieldQA-zh | 50.58 | 50.74 | 49.85 | 50.59 | **52.40** |
| Musique | 24.22 | **24.86** | 24.63 | 24.68 | 24.65 |
| NarrativeQA | 26.56 | 26.63 | 26.17 | **27.36** | 24.54 |
| Passage Count | 1.00 | 1.00 | 1.00 | 1.00 | 0.00 |
| PassageRetrieval-en | 81.00 | 80.50 | 80.00 | 80.50 | **87.00** |
| PassageRetrieval-zh | 62.15 | 58.53 | 54.56 | 61.92 | **62.15** |
| Qasper | 29.21 | 26.00 | 23.63 | 27.02 | **26.93** |
| QMSum | 24.52 | **24.90** | 24.45 | 24.65 | 24.20 |
| RepoBench-P | 38.94 | 38.20 | 37.48 | 38.50 | **46.12** |
| SAMSum | 42.51 | 40.90 | 40.90 | 41.38 | **41.83** |
| TREC | 71.50 | 66.00 | 70.00 | 71.00 | **71.00** |
| TriviaQA | 87.70 | 87.30 | 87.20 | 86.80 | **87.14** |
| VCSUM (zh) | 11.37 | 9.91 | 10.80 | 9.62 | **10.46** |

The score is computed as the ratio of the number of unique words shared between the model's response and the expected answer to the total number of words in the expected answer. Formally, this is given by:

$$\text{score} = \frac{|\text{set}(\texttt{model\_response}) \cap \text{set}(\texttt{expected\_answer})|}{|\text{expected\_answer}|}$$

This approach ensures that the evaluation is robust to minor variations in word order while penalizing the absence of key words from the expected output.

We perform a linear scan over two dimensions: the *insertion depth* of the needle and the *context size* presented to the model. Insertion depth varies across 10 levels: $0\%, 11\%, \ldots, 100\%$ of the corpus length. Context size varies across 13 context sizes as visualized in our paper.

The context provided to the model is formatted as follows:

"<|im_start|> This is a very long story book: <book> {context} </book>.

Based on the content of the book, Question: {retrieval_question}Answer:"

Here, {context} denotes the surrounding text from the haystack corpus, and {retrieval_question} corresponds to the retrieval question.

## A.13 EXPERIMENTS ON QUERY POSITIONING

To further evaluate DuoAttention's robustness compared to SnapKV and PyramidKV, we conducted additional experiments focusing on these methods' dependency on query positioning within the context. Specifically, we designed a scenario in which the query is not positioned at the end of the input context, as SnapKV and PyramidKV typically assume.

In this experiment, the input context was constructed as follows:

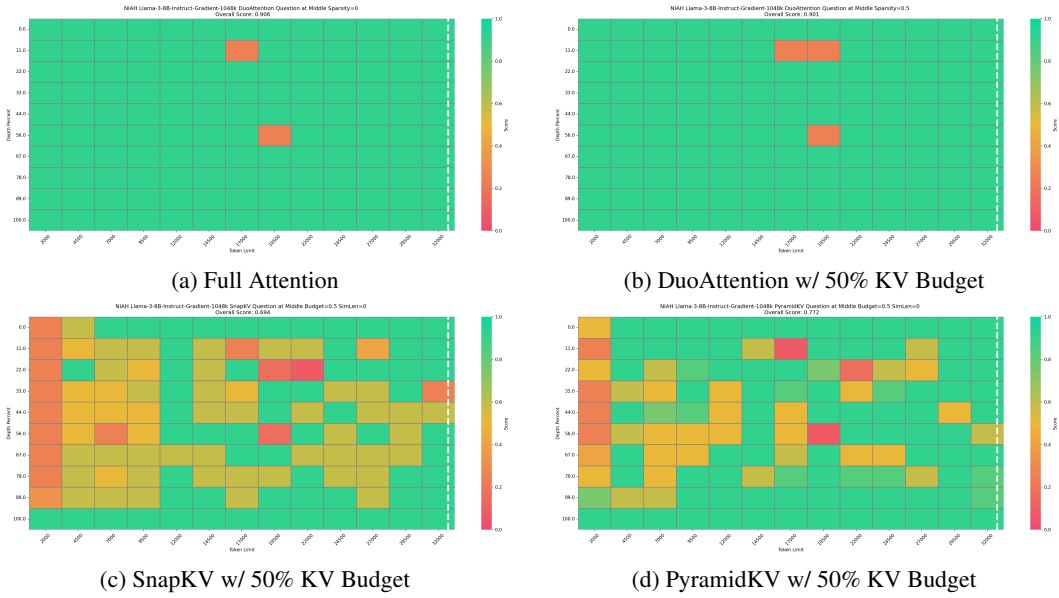

(a) Full Attention            (b) DuoAttention w/ 50% KV Budget

(c) SnapKV w/ 50% KV Budget        (d) PyramidKV w/ 50% KV Budget

Figure 23: NIAH results for Llama-3-8B-Instruct-Gradient-1048k with a 50% KV cache budget. The query of the NIAH benchmark is positioned in the middle of the haystack.

- An instruction was placed at the beginning of the input: *"This is a very long storybook with a question embedded. Please answer the embedded question at the end of the book."*
- The query, *"Q: What is the best thing to do in San Francisco?"*, was positioned immediately before the needle in the middle of the haystack.
- The needle was embedded within the haystack: *"A: The best thing to do in San Francisco is eat a sandwich and sit in Dolores Park on a sunny day."*
- At the end of the context, only a partial answer prompt was provided: *"Answer: The best"* to elicit the model's response.

We evaluated SnapKV, PyramidKV, and DuoAttention on the NIAH benchmark using this context. For this experiment, no simulation of the last tokens was applied; the entire input context (instruction, query, haystack, and partial answer) was provided before KV cache compression.

The results of this experiment are presented in Figure 23. Each subplot illustrates the performance of a method under a 50% KV cache budget. The results reveal several key insights:

1. **SnapKV and PyramidKV Failures:** Both SnapKV and PyramidKV exhibit significant degradation when the query is not at the end of the context. This highlights their reliance on specific assumptions about query locations to guide KV cache pruning. As demonstrated in PyramidKV, even when compressing 32K to 128 with Mistral-7B-Instruct, both SnapKV and PyramidKV exhibit minimal performance degradation. However, this level of performance is only attainable when the query is known and used as observation tokens for pruning. Our updated NIAH results demonstrate that both SnapKV and PyramidKV fail when the observation tokens are not the query tokens, even at a high retention ratio of 50%.

2. **DuoAttention Robustness:** DuoAttention achieves accuracy comparable to full attention in this scenario, underscoring its robustness and general applicability. Unlike SnapKV and PyramidKV, DuoAttention does not rely on the query's position, making it suitable for real-world tasks where query positions are not fixed or predictable.

These findings reinforce the conclusion that DuoAttention offers a more reliable and versatile approach for KV cache compression, particularly in scenarios with diverse query positions.

