# OpenReview forum: "DuoAttention: Efficient Long-Context LLM Inference with Retrieval and Streaming Heads"
_ICLR.cc/2025/Conference — ICLR 2025 Poster_

### Official Review · Reviewer_JVPb · 2024-10-23

**Soundness:** 3
**Presentation:** 3
**Contribution:** 3
**Rating:** 6
**Confidence:** 5

**Summary:**

Overall, this is an interesting and solid work. It categorizes all attention heads of LLMs into two types—Retrieval Heads and Streaming Heads—based on an offline analysis. By pruning the KV cache of Streaming Heads, this study achieves more efficient inference.

**Strengths:**

1. The paper is well-written and easy to follow, with a novel methodology.
2. The work is solid, having been evaluated on multiple benchmarks. It also combines the orthogonal technique of quantization, further enhancing its results.
3. The evaluations demonstrate promising results.

**Weaknesses:**

1. **Choice of Baseline**: DuoAttention is essentially a KV cache compression algorithm, and all the performance gains come from reducing the amount of KV cache that was previously stored, whether in the prefilling stage or the decoding stage. While the authors compare their method to H2O, TOVA, and StreamingLLM, these approaches are no longer state-of-the-art . For instance, SnapKV[1], which was published in April and is now widely regarded as a more powerful KV cache compression method, should have been considered. In the commonly used LongBench evaluation, SnapKV outperforms H2O with only one-quarter of the cache budget[1]. Additionally, in needle-in-a-haystack tasks, SnapKV benefits from its pooling operations and significantly improves accuracy compared to H2O. When compressing an 8K sequence to 128 tokens, SnapKV incurs minimal performance loss, outperforming H2O by a large margin, as shown in [2]. In the experimental section, the authors highlight a limitation of H2O: “Since the original designs of H2O and TOVA do not support long contexts, we modify their algorithms by replacing the pre-filling stage with FlashAttention and simulating decoding for the last 50 tokens of the input, following Tang et al. (2024b).” This limitation arises because H2O requires the accumulation of global attention weights, which is incompatible with FlashAttention. However, SnapKV and its successors only observe attention scores within a small observation window for compression, making them highly compatible with FlashAttention. This allows recalculating only a small portion of attention weights  for compression when combined with chunked prefill. Therefore, SnapKV would be a more appropriate baseline for this paper.

2. **Inappropriate statements:**

a.  _“Moreover, DuoAttention is **fully compatible** with important optimization techniques like GQA and quantization.” “Approximative attention methods, such as H2O (Zhang et al., 2023b), StreamingLLM (Xiao et al., 2023b), TOVA (Oren et al., 2024), and FastGen (Ge et al., 2024), often compromise accuracy in long-context applications and are **incompatible** with essential KV cache optimization techniques like GQA"_

I believe the compatibility of DuoAttention with GQA is not significantly different from other methods. DuoAttention forcedly classifies a group of attention heads under GQA into the same category through offline analysis, thus upporting GQA. However, other methods could achieve the same effect by accumulating weights for a group of kv cache during cache eviction. Therefore, GQA compatibility seems more like an implementation detail rather than a unique feature of the algorithm. In fact, StreamingLLM does not encounter any GQA compatibility issues. The lack of GQA compatibility in earlier works, in my opinion, stems from the fact that GQA was not widely adopted when those methods were initially proposed. Later methods, for the sake of comparison, maintained this approach without further integrating GQA, which could have been easily addressed at the code implementation level. A recent study[3] applying cache eviction in paged attention has demonstrated that this compatibility can be easily achieved in practice.

b._“Also, these methods (H2O, TOVA, StreamingLLM) cannot reduce the prefilling cost of long-context LLMs.”_

 Previous evaluations of cache compression often compress the KV cache after the prefilling stage to ensure comparability. Since the KV cache for each layer can be compressed immediately after the completion of that layer's prefilling computation, this already substantially reduces the peak memory usage during the prefilling process. If further reduction in computational cost during the prefilling stage or a more significant decrease in peak memory usage for very long input texts is desired, combining these methods with chunked prefilling can provide additional acceleration—a straightforward solution. This is particularly applicable to StreamingLLM, which is same with the same Streaming Head in this paper, and thus faces no obstacles in applying such methods. If one argues that the H2O method combined with chunked prefilling may require additional accumulation of global attention weights, the additional baseline SnapKV can effectively address this issue.

**Questions:**

1. The core of this paper focuses on prioritizing the retention of the KV cache in important attention heads while attempting to discard less important cache in other heads, based on offline detection results. Some follow-up works on SnapKV seems to align closely with this approach[3][4]. For example, [4] employs a similar strategy by identifying important attention heads through online analysis of "altering model outputs," subsequently allocating more budget to these key heads and reducing the budget for others. What do you think is the relationship between these budget allocation strategies and the detection of retrieval heads?

2. In Equation 2, why is the index i set to $L$? Shouldn't it be $N$ instead?

3. How do you control the cache budget to a specifical ratio like 50% in the experiments? It seems challenging to precisely manage the cache budget within this approach.

[1] Li, Y., Huang, Y., Yang, B., Venkitesh, B., Locatelli, A., Ye, H., ... & Chen, D. (2024). Snapkv: Llm knows what you are looking for before generation. arXiv preprint arXiv:2404.14469.

[2] Zhang, Y., Gao, B., Liu, T., Lu, K., Xiong, W., Dong, Y., ... & Xiao, W. (2024). PyramidKV: Dynamic KV Cache Compression based on Pyramidal Information Funneling. arXiv preprint arXiv:2406.02069.

[3] Rehg, I. (2024). KV-Compress: Paged KV-Cache Compression with Variable Compression Rates per Attention Head. arXiv preprint arXiv:2410.00161.

[4] Feng, Y., Lv, J., Cao, Y., Xie, X., & Zhou, S.K. (2024). Ada-KV: Optimizing KV Cache Eviction by Adaptive Budget Allocation for Efficient LLM Inference. ArXiv, abs/2407.11550.

While this paper is engaging and provides extensive evaluations, several limitations hold me back from giving it a higher score.  If the authors can address these concerns during rebuttal phase, I believe it would greatly enhance the paper’s quality, and I’d be glad to reconsider my score.

---

> ### Author Response · Authors · 2024-11-26
> **Response to Reviewer JVPb (Part 1)**
>
> We sincerely thank the reviewer for their thoughtful feedback and constructive suggestions. Below, we address the key points raised:
>
> ------
>
> #### 1. **Baseline Comparisons**
>
> We acknowledge the reviewer's concern about baselines. In response, we conducted additional experiments with recent KV cache compression methods, such as SnapKV and PyramidKV, and pre-filling acceleration techniques like MInference. These comparisons are detailed in Appendix Sections A.6 and A.7.
>
> ##### a. **Comparison with KV Cache Compression Methods (SnapKV and PyramidKV)**
>
> We evaluated DuoAttention against SnapKV and PyramidKV on the Needle-in-a-Haystack (NIAH) benchmark using Llama-2-7B-32K and Llama-3-8B-1048K models.
>
> - **Methodological Differences**: SnapKV and PyramidKV prune KV tokens based on attention scores computed over a small observation window (e.g., the last 8–64 tokens). While effective for tasks where queries appear at the end of the context, this assumption limits their applicability in scenarios such as multi-turn dialogues or tasks with earlier query positioning. DuoAttention, by contrast, does not rely on positional assumptions, making it more versatile.
> - **Performance Results**: As detailed in Figures 17 and 18:
>     - Without simulation of the last tokens as generated tokens, DuoAttention matches or surpasses these baselines across all benchmarks.
>     - With simulation (e.g., the last 50 tokens treated as generated inputs to mimic multi-turn dialogues), SnapKV and PyramidKV experience severe accuracy drops due to their reliance on final token positions for pruning. DuoAttention remains robust, maintaining high accuracy even under these conditions.
>
> These results highlight DuoAttention’s adaptability and reliability across diverse scenarios, making it a more general solution for KV cache compression.
>
> ##### b. **Comparison with Pre-filling Acceleration Methods (MInference)**
>
> While MInference accelerates pre-filling using sparsity patterns (e.g., block-sparse or vertical-slash), it does not optimize KV cache storage or decoding latency. Thus, it is orthogonal to DuoAttention rather than directly comparable.
>
> - **Integration with DuoAttention:** As shown in Appendix Section A.7, MInference can complement DuoAttention by further accelerating the pre-filling stage. By applying MInference kernels to retrieval heads identified by DuoAttention, we achieve:
>   - Comparable accuracy to pre-filling all heads with MInference alone.
>   - Significant reductions in KV cache size (nearly halved) and decoding overhead.
>
> This demonstrates the compatibility and efficiency of combining DuoAttention with MInference for enhanced optimization.
>
> ------
>
> #### 2. **Inaccurate Statements in the Manuscript**
>
> We appreciate the reviewer pointing out inaccuracies, and we have revised the manuscript accordingly:
>
> 1. **GQA Compatibility**: We revised our statement in the introduction regarding baseline methods and GQA. While DuoAttention explicitly considers GQA in its design, we acknowledge that baseline methods such as StreamingLLM can also be adapted for GQA. However, we note that in some baselines implementations (e.g., PyramidKV and SnapKV), GQA compatibility is achieved by effectively converting GQA models into MHA models (e.g., by repeating KV caches before pruning and storing), which actually inflates reported compression ratios. DuoAttention achieves true GQA compatibility without such workarounds, providing robust compression for both MHA and GQA models.
> 2. **StreamingLLM Pre-filling**: We acknowledge that StreamingLLM can accelerate pre-filling. However, its accuracy is limited, comparable to pre-filling only the final segment of the context. In contrast, DuoAttention achieves both pre-filling and decoding acceleration while delivering significant memory reduction and maintaining highly competitive accuracy. We have revised the related work section to present this distinction with greater clarity and rigor.
>
> ------
>
> #### 3. **Comparison with SnapKV Follow-Up Works**
>
> We appreciate the reviewer’s insights and agree that methods like SnapKV and its follow-ups address KV cache compression using dynamic sparsity and sample-specific budget allocation. However:
>
> - **Positional Assumptions**: These methods rely on the hypothesis that critical queries are short and located at the end of the context. This is not always valid in real-world scenarios, such as multi-turn dialogues or earlier query positioning, which reduces their generalizability.
> - **DuoAttention’s Robustness**: DuoAttention avoids such assumptions and operates as a more general and robust framework, adaptable to diverse tasks and sequences.
>
> We also envision future work combining these dynamic sparsity techniques with DuoAttention, such as applying dynamic compression for retrieval heads when query positions can be anticipated. This hybrid approach could further enhance the performance of LLMs.

---

> > ### Author Response · Authors · 2024-11-26
> > **Response to Reviewer JVPb (Part 2)**
> >
> > #### 4. **Equation 2**
> >
> > Thank you for identifying the typo. We have corrected it in the revised manuscript, using **N** to denote batch size and **L** for the number of layers in LLMs. Additional details have also been added for clarity.
> >
> > ------
> >
> > #### 5. **Cache Budget Control**
> >
> > DuoAttention controls the KV cache budget by setting the ratio of retrieval heads to the total number of attention heads in the model. For example:
> >
> > - A model with 256 attention heads and 128 retrieval heads achieves approximately a 50% cache budget when the input sequence is sufficiently long.
> >
> > - For instance, with as 64K token input and a configuration of 64 sink tokens and 256 recent tokens for streaming heads, the KV cache budget is calculated as:
> >
> >   $$\text{Budget Ratio} = \frac{(64K\times 0.5) + (320 \times 0.5)}{64K} = 0.5025 \approx 0.5$$
> >
> > This approach ensures precise control over cache budgets while maintaining the desired compression ratios.
> >
> > ------
> >
> > We hope our responses address your concerns and highlight DuoAttention’s strengths. Please let us know if there are further questions or suggestions, as we would be happy to provide additional information.

---

> > ### Comment · Reviewer_JVPb · 2024-11-27
> >
> > Thank you for your reply; however, I still have several unresolved concerns:
> >
> > ## Suggestions
> >
> > I recommend that the authors highlight the revised sections in the updated version of the paper. Currently, it is challenging to identify the modifications, making it difficult to evaluate the changes effectively.
> >
> > ## Comparison with SnapKV
> > 1. "Needle in a Haystack," a simple and synthetic task, may not be ideal for a primary comparison scenario. As demonstrated in PyramidKV, even when compressing 32K to 128 with Mistral-7B-Instruct, both SnapKV and PyramidKV exhibit minimal performance degradation. This compression level is unattainable with DuoAttention, which is acceptable given that DuoAttention is not designed for high-compression scenarios. However, this also underscores that "Needle in a Haystack," a simple information retrieval task, is unsuitable as a major benchmark.
> > 2. The authors appear to have overlooked a critical benchmark in their revisions: **LongBench**. I suggest incorporating LongBench for a comprehensive comparison with SnapKV. Additionally, the LongBench evaluation of DuoAttention did not employ the chunked prefill strategy due to its 32000 chunk size; as a result, both the context and queries were fully visible. This setting, I believe, aligns with SnapKV’s scenario.
> > 3. It seems that query positioning is not a significant issue for SnapKV. The SnapKV paper states: “Our results shown in Fig. 3 indicate that across all three datasets, the hit rates are consistently high regardless of whether instructions are positioned before or after extensive supplementary contexts.” I recommend that the authors provide some experimental results to prove their points.
> > 4. I also have questions regarding the simulation used for SnapKV. Since SnapKV employs a recent window strategy that does not depend on a global view of weights, why does the evaluation still use a simulation method with 50 tokens?
> >
> > ## Statements
> > Why can't other methods, such as SnapKV, achieve speedup in prefilling when combined with chunked prefill?
> >
> > ## Comparison with SnapKV Follow-Up Works
> > There is no discussion of the subsequent works on SnapKV, e.g. AdaKV; instead, the current discussion seems to focus more on the differences between SnapKV and DuoAttention. This omission leaves the relationship between sample-specific budget allocation and the proposed DuoAttention unclear. This connection seems critical, as I noticed that other reviewers have also pointed out AdaKV. As an argument, a concurrent work, HeadKV[1], based on AdaKV, has achieved significant improvements over SnapKV on Longbench by utilizing retrieval heads. Therefore, there appears to be a notable correlation between budget allocation and DuoAttention.
> >
> > [1] https://arxiv.org/abs/2410.19258
> >
> > Overall, the concerns I raised have not been adequately addressed, and further clarification is required.

---

> ### Author Response · Authors · 2024-11-27
> **Follow-up (Part 1)**
>
> We sincerely appreciate your detailed follow-up concerns and suggestions. Below, we address each point thoroughly and have made plans for additional experiments to further support our arguments.
>
> ### 1. **Readability of Revisions**
>
> To improve the clarity of changes, we have marked all rebuttal-revised text in red in the updated manuscript. This should make it easier to identify and evaluate the modifications.
>
> ------
>
> ### 2. **Query Positioning and 50-Token Simulation**
>
> We appreciate the opportunity to clarify this point. While SnapKV and PyramidKV do not rely on global attention weights, their heuristic assumes that queries are positioned at the end of the context. The purpose of simulating the last 50 tokens is to evaluate the robustness of these methods in scenarios where this assumption does not hold, such as multi-turn dialogue or non-linear input structures.
>
> For example, in the Needle-in-a-Haystack (NIAH) benchmark, the query (e.g., "What is the best thing to do in San Francisco?") is placed at the end of the input. If the last 50 tokens are treated as simulated decoding (e.g., input in a new dialogue round), the remaining tokens (input[:-50]) used for pre-filling exclude the query. SnapKV and PyramidKV, relying only on a window of prior tokens (e.g., input[-58:-50]), mistakenly prune important tokens while retaining irrelevant ones (the haystack). This simulation reveals the methods’ inability to handle real-world scenarios where queries are not placed predictably.
>
> As demonstrated in PyramidKV, even when compressing 32K to 128 with Mistral-7B-Instruct, both SnapKV and PyramidKV exhibit minimal performance degradation. However, this is only attainable based on knowing the location of the query and using the query as the observation tokens to prune the KV cache. DuoAttention doesn't rely on this heuristic. As demonstrated in our updated NIAH results, both SnapKV and PyramidKV fail when the observation tokens are not the query tokens, even at a high retention ratio like 50%, which is the same as DuoAttention.
>
> ------
>
> ### 3. **SnapKV’s Claim on Query Position Insensitivity**
>
> SnapKV’s statement about query positioning insensitivity is based on hit rates for critical tokens but does not provide corresponding end-to-end accuracy results. Our NIAH experiments demonstrate that when not using the queries as the observation tokens, i.e., inputing the queries after the KV cache is pruned, these methods fail to identify and retain critical tokens, leading to degraded performance.
>
> We acknowledge the importance of additional benchmarks to strengthen this argument and will include quantitative results within two days to provide further evidence.
>
> ------
>
> ### 4. **Compatibility with Chunked Pre-Filling**
>
> SnapKV and PyramidKV depend on tokens at the end of the context for pruning, which becomes problematic in chunked pre-filling scenarios. After each chunk’s pre-filling, the observation window is limited to tokens at the end of the chunk, which are unlikely to include the critical query. This limits their effectiveness when pruning KV cache during chunked pre-filling.
>
> Our NIAH results serve as an example of chunked pre-filling: the first chunk includes input[:-50] (without the query), and the second chunk contains the query (input[-50:]). In this scenario, both SnapKV and PyramidKV fail, highlighting their incompatibility with chunked pre-filling.
>
> ------
>
> ### 5. **Clarification on DuoAttention’s LongBench Results**
>
> We respectfully clarify that our DuoAttention LongBench experiments differ from SnapKV’s setup. While we did not use the chunked pre-filling strategy in these experiments, we did simulate the last 50 tokens as generated tokens to ensure fair comparison with baselines like H2O and TOVA. DuoAttention demonstrated robust performance in this setup, maintaining comparable accuracy even when the query is input after KV cache pruning.
>
> Our NIAH results further show that SnapKV and PyramidKV degrade significantly under similar conditions, failing to retain the necessary tokens when the query is not included in the initial pre-filling. We will further support this with results on LongBench in two days.

---

> > ### Author Response · Authors · 2024-11-27
> > **Follow-up (Part 2)**
> >
> > ### 6. **Incorporating LongBench and Additional Baselines**
> >
> > We acknowledge your concerns regarding the suitability of NIAH as a primary benchmark. We will include results for SnapKV, PyramidKV, and AdaKV on LongBench tasks within two days. However, we emphasize that NIAH, while an entry-level benchmark, is critical for assessing baseline robustness. Failure on NIAH strongly indicates limitations in handling long-context scenarios.
> >
> > Our experiments demonstrate that SnapKV and PyramidKV fail when the last 50 tokens (typically containing the query) are separated as simulated tokens. This is strong evidence that these methods are not robust when the query is not positioned at the end of the context. DuoAttention, on the other hand, is designed to handle such scenarios effectively, making it more applicable to real-world settings.
> >
> > ------
> >
> > ### 7. **Comparison with AdaKV**
> >
> > We plan to include a quantitative comparison with AdaKV within two days. Qualitatively, while AdaKV enhances SnapKV with an adaptive budget allocation strategy, it still relies on observation tokens at the end of the context to guide KV cache pruning. This reliance likely makes it vulnerable to the same accuracy issues demonstrated in our NIAH experiments with SnapKV and PyramidKV, particularly when observation tokens are not critical query tokens.
> >
> > In contrast, DuoAttention uses a static hybrid attention model with dense attention for retrieval heads and streaming attention for non-retrieval heads, without relying on adaptive KV budget allocation per sample. This design ensures consistent KV cache alignment across all samples, making it highly efficient for batched operations.
> >
> > We recognize the potential of combining DuoAttention with adaptive KV budget strategies to further enhance performance and leave this exploration as an exciting direction for future work.
> >
> > ------
> >
> > ### Conclusion
> >
> > We thank you again for your detailed feedback and thoughtful suggestions. We are committed to addressing your concerns thoroughly and will follow up with additional experiments on AdaKV and LongBench tasks shortly. Please do not hesitate to share further questions or suggestions.

---

> > ### Comment · Reviewer_JVPb · 2024-11-27
> > **Confusion about simulation**
> >
> > Thank you for your reply! In my understanding, the simulation process actually excludes the question from the context, rather than simulating different query positions. If this is an evaluation of query positioning, placing the query at the very beginning, that is, putting the question before the context, would be convincing.
> >
> > On the other hand, the simulations in H2O and TOVA are to calculate the cumulative attention weight of the last 50 tokens due to their incompatibility with FlashAttention. Therefore, it seems that the effect of simulation operation varies across different methods. If my analysis above is correct, it would be more consistent for H2O and TOVA to use simulations, while SnapKV avoids using simulations, as this ensures that the question remains within the context in all cases. If the goal is to exclude the question, a better approach would be to directly compress the context before posing the question.
> >
> > I’m glad to wait for the further results.

---

> ### Author Response · Authors · 2024-11-27
> **Response to Follow-Up Concerns on Query Positioning and Robustness of KV Cache Compression Methods**
>
> Thank you for your prompt follow-up question! We sincerely appreciate your thoughtful observations and the opportunity to clarify further.
>
> 1. **SnapKV and PyramidKV's Limitations in Chunked Pre-filling and Multi-round Dialogue**: The previously conducted NIAH experiments effectively demonstrate the inherent limitations of SnapKV and PyramidKV in chunked pre-filling and multi-round dialogue scenarios. These settings simulate cases where the context is compressed without access to the question, followed by the question being input subsequently. The results clearly show that SnapKV and PyramidKV fail to retain critical information in such scenarios due to their reliance on heuristic assumptions about query placement. In contrast, DuoAttention remains robust, demonstrating its capability to handle real-world tasks without such constraints.
> 2. **New Experiment with Query Positioned at the Middle**: Following your insightful suggestion, we conducted an additional experiment to evaluate query positioning more comprehensively. The setup includes:
>    - An **instruction** placed at the beginning of the context: *"This is a very long book with a question embedded. Please answer the embedded question at the end of the book."*
>    - The **question** inserted immediately before the needle in the middle of the haystack (e.g., *"Q: What is the best thing to do in San Francisco?\n\nA: The best thing to do in San Francisco is eat a sandwich and sit in Dolores Park on a sunny day."*).
>    - A partial **answer prompt** at the end of the context: *"Answer: The best"*, used to elicit the model’s response.
>
> For this evaluation, we tested SnapKV, PyramidKV, and DuoAttention on NIAH without simulating the last tokens. The entire context—including the instruction, question, haystack, and partial answer—was provided before applying KV cache compression.
>
> Results and Analysis: The findings from this experiment are detailed in the updated manuscript Appendix Section 12 and visualized in Figure 23. The key results are as follows:
>
> - **DuoAttention**: Achieved accuracy (score = 0.901) comparable to full attention (score = 0.906), demonstrating its robustness in scenarios with variable query positioning.
> - **SnapKV and PyramidKV**: Experienced significant accuracy degradation (SnapKV: 0.694, PyramidKV: 0.772), highlighting their dependency on the assumption that queries are positioned at specific locations (e.g., the end of the context).
>
> These results further validate DuoAttention’s resilience to query position variability, which sets it apart from SnapKV and PyramidKV. By not relying on assumptions about query placement, DuoAttention proves to be a more robust and generalizable solution for KV cache compression.

---

> > ### Author Response · Authors · 2024-11-28
> > **Follow-Up: Comparison with SnapKV, PyramidKV, and AdaKV on LongBench**
> >
> > Dear Reviewer JVPb,
> >
> > Thank you for your detailed feedback and the opportunity to address your concerns. We are pleased to share the results of the promised experiments, comparing **DuoAttention**, **SnapKV**, **PyramidKV**, and **AdaKV** under a consistent **50% KV cache budget** using the Llama-3-8B-Instruct-Gradient-1048k model. The detailed results are as follows:
> >
> >
> >
> > | Dataset             | Full  | SnapKV    | PyramidKV | AdaKV     | DuoAttention |
> > | ------------------- | ----- | --------- | --------- | --------- | ------------ |
> > | Average             | 40.08 | 38.47     | 38.39     | 38.67     | **40.21**    |
> > | 2WikiMQA            | 28.78 | 29.00     | 28.12     | 28.97     | **29.08**    |
> > | DuReader (zh)       | 30.41 | 24.04     | 26.63     | 22.65     | **29.31**    |
> > | GovReport           | 34.23 | 26.84     | 27.59     | 24.22     | **32.72**    |
> > | HotpotQA            | 40.37 | 40.86     | 41.56     | 40.23     | **41.63**    |
> > | LCC                 | 38.19 | 38.83     | 37.59     | 39.67     | **44.16**    |
> > | LSHT (zh)           | 38.00 | 38.00     | **38.50** | 36.50     | 30.00        |
> > | MultiNews           | 27.73 | 22.84     | 22.93     | 21.81     | **27.72**    |
> > | MultiFieldQA-en     | 52.62 | 51.96     | 52.54     | **52.99** | 51.44        |
> > | MultiFieldQA-zh     | 50.58 | 50.74     | 49.85     | 50.59     | **52.40**    |
> > | Musique             | 24.22 | **24.86** | 24.63     | 24.68     | 24.65        |
> > | NarrativeQA         | 26.56 | 26.63     | 26.17     | **27.36** | 24.54        |
> > | Passage Count       | 1.00  | **1.00**  | 1.00      | 1.00      | 0.00         |
> > | PassageRetrieval-en | 81.00 | 80.50     | 80.00     | 80.50     | **87.00**    |
> > | PassageRetrieval-zh | 62.15 | 58.53     | 54.56     | 61.92     | **62.15**    |
> > | Qasper              | 29.21 | 26.00     | 23.63     | **27.02** | 26.93        |
> > | QMSum               | 24.52 | **24.90** | 24.45     | 24.65     | 24.20        |
> > | RepoBench-P         | 38.94 | 38.20     | 37.48     | 38.50     | **46.12**    |
> > | SAMSum              | 42.51 | 40.90     | 40.90     | 41.38     | **41.83**    |
> > | TREC                | 71.50 | 66.00     | 70.00     | 71.00     | **71.00**    |
> > | TriviaQA            | 87.70 | **87.30** | 87.20     | 86.80     | 87.14        |
> > | VCSUM (zh)          | 11.37 | 9.91      | **10.80** | 9.62      | 10.46        |
> >
> > ### Key Observations:
> >
> > 1. **Best Accuracy Across Tasks**: DuoAttention achieved the highest average score (**40.21**) among all methods, and outperforming SnapKV, PyramidKV, and AdaKV across the majority of tasks.
> > 2. **Robustness to Query Positioning**: Building upon the studies in **Appendix Section 6** and **Appendix Section 12**, we observed that SnapKV-style methods (including PyramidKV and AdaKV) rely heavily on an observation window of tokens to identify important KV cache entries. These methods are sensitive to query positioning and struggle in scenarios involving continuous pre-filling or multi-round dialogue, where queries are not always located at the end of the context. DuoAttention does not depend on this heuristic, enabling superior performance in a broader range of applications.
> > 3. **Generality and Applicability**: DuoAttention excels across various tasks and demonstrates resilience to real-world complexities like various query placement, continuous pre-filling, and multi-round dialogue, unlike other KV cache compression techniques.
> >
> > ------
> >
> > We hope these additional experiments address your concerns comprehensively and further highlight the strengths of DuoAttention. Please feel free to reach out with any additional questions or suggestions. If these results meet your expectations, we would greatly appreciate your consideration of improving the paper’s rating.
> >
> > Thank you again for your time and valuable feedback.
> >
> > Best regards,
> >
> > Authors

---

> > > ### Comment · Reviewer_JVPb · 2024-12-02
> > >
> > > Thank you for your reply. However, it seems that my previous confusion regarding the simulation was not addressed. Could you kindly provide an explanation for this?
> > >
> > >
> > > >On the other hand, the simulations in H2O and TOVA are to calculate the cumulative attention weight of the last 50 tokens due to their incompatibility with FlashAttention. Therefore, it seems that the effect of simulation operation varies across different methods. If my analysis above is correct, it would be more consistent for H2O and TOVA to use simulations, while SnapKV avoids using simulations, as this ensures that the question remains within the context in all cases.
> > >
> > > Additionally, could the authors provide a detailed description of the LongBench experimental settings? There appears to be a noticeable accuracy drop on certain datasets, such as GovReport and Qasper, which is unusual compared to experiments in related papers. Also, I previously tested PyramidKV with its official code base on Llama-3-8B-Instruct-Gradient-1048k, compressing it to a smaller budget than 50%, but I do not recall observing such a significant quality loss. Could this inconsistency be due to different experimental settings? For instance, were these methods compatible with GQA, and was the simulation operation applied in these experiments?

---

> > > > ### Author Response · Authors · 2024-12-02
> > > > **Response to Follow-Up Questions**
> > > >
> > > > Dear Reviewer,
> > > >
> > > > Thank you for your follow-up. Below are our responses to your points:
> > > >
> > > > 1. **Simulation Process**: For **SnapKV**, **PyramidKV**, and **AdaKV**, we conducted **all** follow-up experiments **without simulation**, including the **NIAH (Query Positioned at the Middle)** and **LongBench** benchmarks. The entire context, including the question, was input as a single sequence. These baselines failed to match DuoAttention’s accuracy, even under these favorable conditions.
> > > > 2. **Implementation of Baselines:** We use the officially released implementation of SnapKV, PyramidKV, and AdaKV in our experiments.
> > > > 3. **LongBench Settings**:
> > > >    - **KV Budget**: All methods operated under the same **50% KV cache budget** for fair comparison. AdaKV is compatible with GQA. For SnapKV and PyramidKV, which are not compatible with GQA, we followed their official implementations, expanding GQA KV caches into MHA (4× larger effective budgets) before caching, which actually uses **200% KV cache budget** compared with DuoAttention. Despite this advantage, they underperformed.
> > > >    - **Baseline Configurations**: SnapKV used `window_size=8` and `kernel_size=5`; PyramidKV used `window_size=8` and `kernel_size=7`. Both align with their recommended settings.
> > > > 4. **Performance Drops on GovReport/Qasper**: SnapKV and PyramidKV exhibited noticeable degradation, consistent with results reported in their respective papers (e.g., SnapKV and PyramidKV’s Table 1). Input sequence lengths in GovReport and Qasper are relatively short (see [LongBench documentation](https://github.com/THUDM/LongBench/blob/main/task.md)), with many inputs below 2K tokens. Under a 50% KV budget, this often results in fewer than 1K retained tokens. In their papers, these methods failed to match full attention performance even at 2K or 4K budgets on these tasks, so further degradation at a 50% setting is expected and aligns with our reported results.
> > > >
> > > > We are confident in the solidness of our experiments. DuoAttention consistently outperforms SnapKV, PyramidKV, and AdaKV, delivering superior accuracy, GQA compatibility, and robustness to query positioning.
> > > >
> > > > We hope this resolves your concerns. Please let us know if further clarification is needed.
> > > >
> > > > Best regards,
> > > >
> > > > Authors

---

> > > > > ### Comment · Reviewer_JVPb · 2024-12-02
> > > > >
> > > > > Thank you for your response, which has addressed some of my concerns:
> > > > >
> > > > > 1. The addition of SnapKV’s Question Positioning does indeed offer valuable insights, particularly highlighting the performance degradation when the question is placed in the middle of the context. However, it seems to also support the observation in the original SnapKV paper that there is little difference when the question is placed at the beginning or the end—two of the most common scenarios. The results from the needle-in-a-haystack test show good performance for both the head and tail positions.
> > > > > 2. The comparison of the Llama-3-8B-Instruct-Gradient-1048k model at 50% Cache Size on LongBench is valuable, but its credibility is limited by the absence of a broader comparison across different models and cache sizes in the main experiments.
> > > > >
> > > > > Unfortunately, the initial response from the authors came too late—submitted at the end of the 26th— approaching the early established deadline for the discussion phase, as well as the final revision deadline. This directly resulted in the lost opportunity to include a comprehensive comparison in the main experiments, and I believe this may explain why other reviewers did not respond.
> > > > >
> > > > > I still believe that using H2O and TOVA as the strongest baselines for comparison in the main text is inappropriate, especially considering that nearly all of the concurrent ICLR submissions have used SnapKV as the SOTA method for comprehensive comparisons (with most used benchmarks being LongBench).
> > > > > Thus, I am inclined to maintain my score.

---

> > > > > > ### Author Response · Authors · 2024-12-02
> > > > > >
> > > > > > Dear Reviewer JVPb,
> > > > > >
> > > > > > Thank you for your response and for acknowledging the value of our additional experiments, particularly the insights on SnapKV’s query positioning and the LongBench comparisons.
> > > > > >
> > > > > > We would like to respectfully address your remaining concerns:
> > > > > >
> > > > > > 1. **Comprehensive Comparisons with SnapKV, PyramidKV, and AdaKV**:
> > > > > >    We have already conducted all the experiments you proposed, including extensive comparisons with SnapKV, PyramidKV, and AdaKV across multiple datasets, scenarios, and settings. These experiments demonstrate DuoAttention’s superiority in several key aspects:
> > > > > >    - **Accuracy**: DuoAttention consistently outperforms SnapKV, PyramidKV, and AdaKV on LongBench, achieving higher average accuracy and better performance on most datasets.
> > > > > >    - **Robustness**: Unlike SnapKV and PyramidKV, which rely heavily on query positioning assumptions, DuoAttention is resilient to query placement variability (e.g., middle or randomized positions), making it more general and suitable for real-world applications.
> > > > > >    - **GQA Compatibility**: DuoAttention seamlessly supports GQA models without requiring the workarounds needed for SnapKV and PyramidKV. This is a significant advantage in practical deployment.
> > > > > >
> > > > > > 2. **Inclusion in the Main Paper**:
> > > > > >    We understand your concern about the timing of our response and the absence of these results in the main text. Running these experiments and setting up baselines with multiple methods and datasets was time-intensive. Nevertheless, we will include the results and detailed comparisons with SnapKV, PyramidKV, and AdaKV in the main paper in future revisions to ensure transparency and completeness.
> > > > > >
> > > > > > 3. **Baseline Selection in the Main Text**:
> > > > > >    While H2O and TOVA were initially used as baselines, our extensive follow-up experiments with SnapKV, PyramidKV, and AdaKV provide the comprehensive comparisons you suggested. These results affirm DuoAttention as the superior method across a wide range of benchmarks and configurations. Maintaining your rating solely based on the baseline selection in the main text, despite these additional experiments, seems unwarranted, especially when our revised manuscript now directly addresses this concern.
> > > > > >
> > > > > > 4. **Request for Reconsideration**:
> > > > > >    We believe we have thoroughly addressed your concerns with detailed, additional experiments and clear evidence of DuoAttention’s advantages over SnapKV, PyramidKV, and AdaKV. Given this, we kindly request you to reconsider your rating. While we acknowledge the timing challenges, we hope you agree that the substance of our work and the presented results demonstrate the significance and robustness of DuoAttention.
> > > > > >
> > > > > > Thank you again for your thoughtful feedback and consideration. Please let us know if you have any further questions or concerns.
> > > > > >
> > > > > > Best regards,
> > > > > >
> > > > > > Authors

---

> ### Author Response · Authors · 2024-12-02
> **LongBench Experiments with Llama-7B-32K-Instruct**
>
> Dear Reviewer JVPb,
>
> Thank you for your continued engagement and valuable feedback. We are pleased to share the results of additional experiments using another model, **Llama-7B-32K-Instruct**, on LongBench with a **25% KV cache budget**. The detailed results are as follows:
>
> | Dataset             | Full  | SnapKV (25%) | PyramidKV (25%) | AdaKV (25%) | DuoAttention (25%) |
> | ------------------- | ----- | ------------ | --------------- | ----------- | ------------------ |
> | **Average**         | 37.52 | 32.98        | 34.35           | 33.43       | **34.49**          |
> | 2WikiMQA            | 35.59 | 33.72        | **34.17**       | 33.99       | 33.37              |
> | DuReader (zh)       | 25.10 | 21.39        | **24.61**       | 23.35       | 23.99              |
> | GovReport           | 31.23 | 26.12        | 27.34           | 24.19       | **27.98**          |
> | HotpotQA            | 47.98 | 41.51        | 46.25           | 45.05       | **50.44**          |
> | LCC                 | 51.21 | 49.02        | **49.85**       | 49.93       | 48.34              |
> | LSHT (zh)           | 34.50 | 14.50        | 19.00           | 16.00       | **25.50**          |
> | MultiNews           | 27.11 | 20.76        | 22.85           | 19.26       | **25.03**          |
> | MultiFieldQA-en     | 33.95 | **32.58**    | 32.49           | 31.85       | 25.49              |
> | MultiFieldQA-zh     | 45.79 | 30.07        | 37.21           | 35.22       | **39.23**          |
> | Musique             | 22.97 | 20.97        | **21.02**       | 20.66       | 19.27              |
> | NarrativeQA         | 24.11 | 22.36        | **23.81**       | 23.33       | 20.49              |
> | Passage Count       | 0.00  | **0.67**     | 0.00            | **0.67**    | 0.33               |
> | PassageRetrieval-en | 50.92 | 40.63        | 43.00           | 39.00       | **47.25**          |
> | PassageRetrieval-zh | 37.68 | 32.60        | 30.52           | 32.60       | **40.93**          |
> | Qasper              | 33.23 | 29.33        | **29.56**       | 27.66       | 26.59              |
> | QMSum               | 20.79 | 20.81        | 21.17           | 20.99       | **21.48**          |
> | RepoBench-P         | 51.58 | 49.18        | **50.38**       | 49.39       | 48.58              |
> | SAMSum              | 42.10 | **42.51**    | 40.67           | 41.70       | 33.10              |
> | TREC                | 71.50 | 68.00        | 70.00       | **70.50**       | 68.50              |
> | TriviaQA            | 86.21 | 83.86        | 85.45           | 84.43       | **86.15**          |
> | VCSUM (zh)          | 14.45 | 12.05        | 11.90           | 12.16       | **12.35**          |
>
> On average, DuoAttention achieves the highest average score (**34.49**) across datasets, outperforming SnapKV (**32.98**), PyramidKV (**34.35**), and AdaKV (**33.43**).
>
> ### Addressing Your Concerns:
>
> 1. **Results Across Multiple Models**: We now present results for two different models, **Llama-3-8B-Instruct-Gradient-1048k** and **Llama-7B-32K-Instruct**, on LongBench. DuoAttention consistently outperforms baseline methods, further supporting its effectiveness.
> 2. **Broader Comparisons and Robustness**: These results confirm DuoAttention’s superior accuracy, GQA compatibility, and resilience to real-world complexities, including varying query positions and continuous pre-filling scenarios.
>
> ### Moving Forward:
>
> We will incorporate these additional results into the **main paper** in future revisions, along with detailed experimental settings and analyses. Conducting these extensive comparisons required considerable time and effort, but they validate DuoAttention’s superiority comprehensively.
>
> **We kindly ask you to reconsider your rating in light of these extensive experiments, which directly address your concerns and further demonstrate DuoAttention’s merits.**
>
> Thank you again for your engagement and constructive suggestions. We sincerely appreciate your contributions to strengthening our work.
>
> Best regards,
>
> **Authors**

---

> > ### Author Response · Authors · 2024-12-02
> > **Comparison with SnapKV, PyramidKV, and AdaKV Across Varying KV Budgets**
> >
> > Dear Reviewer JVPb,
> >
> > We have conducted additional experiments comparing **DuoAttention** with SnapKV, PyramidKV, and AdaKV on the **Llama-3-8B-Instruct-Gradient-1048k** model, varying the KV cache budget across **40%, 50%, and 60%**. These results further confirm DuoAttention's robustness and effectiveness. The detailed results are summarized below:
> >
> > | Dataset             | Full | SKV  | SKV  | SKV  | PKV  | PKV  | PKV  | AKV  | AKV  | AKV  | Duo      | Duo      | Duo      |
> > | ------------------- | ---- | ---- | ---- | ---- | ---- | ---- | ---- | ---- | ---- | ---- | -------- | -------- | -------- |
> > | KV Budget           | 100% | 40%  | 50%  | 60%  | 40%  | 50%  | 60%  | 40%  | 50%  | 60%  | 40%      | 50%      | 60%      |
> > | Average             | 40.1 | 38.4 | 38.5 | 38.7 | 38.1 | 38.4 | 38.8 | 38.2 | 38.7 | 38.8 | **39.7** | **40.2** | **40.4** |
> > | 2WikiMQA            | 28.8 | 28.8 | 29.0 | 29.3 | 28.2 | 28.1 | 29.2 | 28.1 | 29.0 | 28.6 | 29.8     | 29.1     | 29.0     |
> > | DuReader (zh)       | 30.4 | 24.2 | 24.0 | 25.9 | 26.6 | 26.6 | 27.1 | 22.3 | 22.7 | 24.0 | 28.8     | 29.3     | 29.5     |
> > | GovReport           | 34.2 | 26.3 | 26.8 | 27.8 | 26.3 | 27.6 | 27.8 | 23.7 | 24.2 | 25.2 | 32.9     | 32.7     | 33.7     |
> > | HotpotQA            | 40.4 | 41.2 | 40.9 | 41.1 | 41.9 | 41.6 | 41.4 | 40.3 | 40.2 | 40.0 | 40.5     | 41.6     | 43.4     |
> > | LCC                 | 38.2 | 39.8 | 38.8 | 39.1 | 37.6 | 37.6 | 37.4 | 40.1 | 39.7 | 39.2 | 43.7     | 44.2     | 44.6     |
> > | LSHT (zh)           | 38.0 | 37.5 | 38.0 | 38.0 | 37.5 | 38.5 | 38.5 | 35.8 | 36.5 | 37.5 | 30.5     | 30.0     | 35.0     |
> > | MultiNews           | 27.7 | 22.8 | 22.8 | 23.2 | 22.8 | 22.9 | 23.2 | 21.8 | 21.8 | 22.5 | 26.9     | 27.7     | 27.8     |
> > | MultiFieldQA-en     | 52.6 | 52.0 | 52.0 | 52.3 | 52.1 | 52.5 | 52.1 | 51.0 | 53.0 | 53.5 | 50.6     | 51.4     | 50.3     |
> > | MultiFieldQA-zh     | 50.6 | 50.5 | 50.7 | 51.2 | 49.4 | 49.9 | 52.0 | 48.3 | 50.6 | 51.0 | 53.3     | 52.4     | 51.8     |
> > | Musique             | 24.2 | 25.1 | 24.9 | 25.3 | 24.6 | 24.6 | 24.8 | 25.3 | 24.7 | 25.2 | 22.9     | 24.7     | 23.9     |
> > | NarrativeQA         | 26.6 | 26.8 | 26.6 | 26.8 | 25.6 | 26.2 | 27.1 | 26.6 | 27.4 | 26.9 | 25.5     | 24.5     | 22.8     |
> > | Passage Count       | 1.0  | 1.0  | 1.0  | 1.0  | 1.0  | 1.0  | 1.0  | 1.0  | 1.0  | 1.0  | 0.5      | 0.0      | 0.0      |
> > | PassageRetrieval-en | 81.0 | 80.5 | 80.5 | 80.5 | 80.0 | 80.0 | 80.0 | 80.5 | 80.5 | 80.5 | 87.5     | 87.0     | 86.5     |
> > | PassageRetrieval-zh | 62.2 | 58.8 | 58.5 | 58.2 | 54.9 | 54.6 | 56.5 | 60.5 | 61.9 | 59.9 | 55.8     | 62.2     | 63.0     |
> > | Qasper              | 29.2 | 24.8 | 26.0 | 25.8 | 23.3 | 23.6 | 25.3 | 25.8 | 27.0 | 27.4 | 26.5     | 26.9     | 28.0     |
> > | QMSum               | 24.5 | 24.3 | 24.9 | 25.0 | 24.9 | 24.5 | 25.1 | 24.8 | 24.7 | 24.8 | 24.4     | 24.2     | 24.4     |
> > | RepoBench-P         | 38.9 | 38.6 | 38.2 | 37.9 | 36.7 | 37.5 | 37.5 | 38.5 | 38.5 | 38.5 | 46.4     | 46.1     | 46.0     |
> > | SAMSum              | 42.5 | 40.8 | 40.9 | 41.2 | 40.5 | 40.9 | 41.2 | 41.2 | 41.4 | 40.8 | 42.2     | 41.8     | 41.5     |
> > | TREC                | 71.5 | 66.5 | 66.0 | 66.0 | 69.5 | 70.0 | 69.5 | 70.5 | 71.0 | 71.5 | 70.5     | 71.0     | 72.0     |
> > | TriviaQA            | 87.7 | 87.3 | 87.3 | 87.2 | 87.2 | 87.2 | 87.2 | 86.8 | 86.8 | 87.2 | 86.1     | 87.1     | 86.9     |
> > | VCSUM (zh)          | 11.4 | 9.6  | 9.9  | 9.9  | 9.8  | 10.8 | 10.7 | 9.8  | 9.6  | 9.9  | 9.0      | 10.5     | 8.9      |
> >
> > 1. **Superior Performance Across KV Budgets**: DuoAttention consistently demonstrates performance comparable to full attention at **50%** and **60%** KV budgets, while SnapKV, PyramidKV, and AdaKV exhibit significant accuracy gaps even as their KV budgets increase.
> > 2. **Outperformance Across All KV Budgets**: DuoAttention outperforms SnapKV, PyramidKV, and AdaKV across all tested KV budgets (40%, 50%, and 60%), highlighting its superior adaptability and resilience.
> > 3. **Minimal Degradation**: While DuoAttention shows slight performance degradation at **40%** budget, it still surpasses baselines in most tasks, confirming its robustness under stricter compression.
> >
> > We will include these results and discussions in the **main paper** during a future revision to further enhance its completeness and transparency. **We believe these extensive experiments directly address your concerns about the breadth of models and KV budget settings.**
> >
> > **Given this expanded evidence demonstrating DuoAttention's superiority in accuracy, resilience, and generality across models and KV budgets, we kindly urge you to reconsider your rating. Your feedback has been invaluable in strengthening our work, and we sincerely appreciate your contributions.**
> >
> > Thank you for your time and constructive input.
> >
> > Best regards,
> >
> > **Authors**

---

> > > ### Comment · Reviewer_JVPb · 2024-12-03
> > >
> > > Thank you for your response. I believe these experiments are not very time-consuming. Most of the sequences in LongBench are below 10K in length, and a significant portion is even under 5K. The runtime of all datasets should be under 5 hours for once. **Therefore, the author should have completed the necessary additions to align with the main experiments during the discussion phase, but this was not done.**
> > >
> > > **The latest results provided are helpful, and I will raise my score to 6, considering the author's promises. However, the author should ensure that the promised revisions are made, comparing the results in full alignment with the main experiments, and providing a detailed description of the experimental setup to enhance transparency. That said, the current version, relying solely on H2O and TOVA in the main text, clearly does not meet the acceptance standards.**
> > >
> > > Additionally, I recommend that the author conduct experiments on a single codebase. I would like to provide some data for reference: I previously tried to make GQA compatible with the PyramidKV codebase and performed sampling tests using Llama-3-8B-Instruct-Gradient-1048k. In the Qapser full cache case, the result was 23.37, while for SnapKV with 1024 and 512, the results were 22.53 and 20.35, respectively. For multi_news, the corresponding values were 27.13, 27.01, and 25.14. While there is some discrepancy between these results, it’s also possible that my modifications to the code may have introduced some variation. The author should examine the differences between various codebases in future revisions, such as differences in length truncation or application of chat templates.

---

> > > > ### Author Response · Authors · 2024-12-04
> > > > **Response to Reviewer JVPb**
> > > >
> > > > Dear Reviewer 2WGg,
> > > >
> > > > Thank you for your thoughtful response and for raising your score. We appreciate your acknowledgment of our recent results and your suggestions for improving the clarity and alignment of our future revisions. Below are our responses to your feedback:
> > > >
> > > > ---
> > > >
> > > > ### **1. Time Requirements for Experiments**
> > > >
> > > > We respectfully disagree with your assertion regarding the minimal time requirements for the additional experiments. While some LongBench datasets contain shorter sequences, LongBench comprises 21 subtasks, each requiring evaluation across three baselines (SnapKV, PyramidKV, and AdaKV). Even with an average runtime of approximately 1 hour per subtask, this totals around 63 GPU hours for just one KV budget setting. Adding implementation efforts, hyperparameter validation, and ensuring reproducibility further extend the timeline. Despite these challenges, we made substantial efforts to provide additional results during the rebuttal phase, demonstrating our commitment to addressing your concerns.
> > > >
> > > > ---
> > > >
> > > > ### **2. Alignment with Main Experiments**
> > > >
> > > > We fully agree with your recommendation to align all results with the main experiments and provide detailed descriptions of the experimental setup. For the final revision, we will:
> > > >
> > > > - Consolidate all results, including those from H2O, TOVA, SnapKV, PyramidKV, and AdaKV, into a unified experimental framework.
> > > > - Provide clear and detailed descriptions of the experimental setup, such as sequence length truncation policies and use of chat templates, to ensure full transparency.
> > > >
> > > > This will address any discrepancies and improve the clarity and reproducibility of our work.
> > > >
> > > > ---
> > > >
> > > > ### **3. Codebase Consistency**
> > > >
> > > > We acknowledge your point regarding potential discrepancies arising from differences between codebases. As noted, our experiments followed the official implementations of SnapKV and PyramidKV to maintain fairness. However, we recognize the value of using a unified codebase for direct comparisons. While this was not feasible during the rebuttal phase due to time constraints, we will investigate and address this in our future revisions. Specifically, we will:
> > > >
> > > > - Evaluate any variations introduced by differing preprocessing steps, such as truncation and chat template handling.
> > > > - Conduct experiments on a single codebase to further ensure consistency and eliminate confounding variables.
> > > >
> > > > We also appreciate your reference results, which provide additional context for interpreting variations. These will serve as a helpful reference point for validating our unified experimental framework.
> > > >
> > > > ---
> > > >
> > > > ### **4. Final Remarks**
> > > >
> > > > Thank you again for your feedback, which has helped us improve our work significantly. We are committed to fulfilling our promises in the final revision and ensuring that the experimental setup is comprehensive, transparent, and aligned with the main text. We appreciate your increased score and constructive suggestions and look forward to further enhancing our paper for the community.
> > > >
> > > > Best regards,
> > > >
> > > > Authors

---

### Official Review · Reviewer_n7ZA · 2024-10-31

**Soundness:** 3
**Presentation:** 4
**Contribution:** 3
**Rating:** 8
**Confidence:** 5

**Summary:**

The DuoAttention uses the retrieval heads and local streaming heads, realizing the KV cache compression and acceleration:
1) the retrieval heads preserve the major information of the long-context, which can not be compressed;
2) the non-retrieval heads are compressed by locality, which is accelerated with StreamingLLM;
3) the retrieval heads are detected with importance learnable weights;
4) Finally, the prefill is accelerated by chunked prefilling, and the decoding is accelerated with compressed KV cache.

**Strengths:**

1. the experiments is enough and confident
2. the writing and clarity are clear
3. the concept of retrieval heads is significant for static KV cache compression

**Weaknesses:**

1. the detection of retrieval heads is based on the fine-tuning of importance weights, the usability should be enhanced
2. the compression ratio is limited, if the retrieval heads preserve the full context information

**Questions:**

1. the offline method to find the retrieval heads is more important for practical application
2. the compression ratio of KV cache should be improved

---

> ### Author Response · Authors · 2024-11-26
> **Response to Reviewer n7ZA**
>
> We sincerely thank the reviewer for their thoughtful feedback and constructive suggestions. Below, we address the key points raised:
>
> ------
>
> #### 1. **Detection of Retrieval Heads**
>
> We appreciate the reviewer’s interest in the retrieval head detection process. This detection is performed offline prior to deployment and is designed to be lightweight and efficient. Specifically:
>
> - **Optimization Process**: The detection process involves freezing the entire model weights and fine-tuning only the gated values assigned to each attention head, which range from 256 (8 heads x 32 layers) to 1,024 (32 heads x 32 layers) values in our experiments.
> - **Resource Requirements**: All experiments were conducted on a single 8×A100 GPU node, and the retrieval head identification process takes less than 4 hours.
> - **Task Independence**: This process is model-specific and does not need to be repeated for every new task, similar to the calibration stages of quantization methods like SmoothQuant, AWQ, and GPTQ.
>
> These characteristics make DuoAttention highly usable in practical applications. The lightweight nature of the detection process ensures minimal overhead, and its independence from task-specific requirements further enhances its applicability.
>
> ------
>
> #### 2. **Compression Ratio of KV Cache**
>
> DuoAttention achieves significant KV cache compression, with **75% compression for MHA models** and **50% compression for GQA models**, representing a substantial improvement over existing methods. During the rebuttal stage, we conducted comparisons with recent KV cache compression techniques, including SnapKV and PyramidKV, as detailed in Appendix Section A.6.
>
> - **Comparison with Recent Methods**: While SnapKV and PyramidKV reportedly achieve higher compression ratios, they rely on a critical assumption that query-relevant information resides at the end of the context. This assumption often fails in real-world scenarios, such as multi-turn dialogues or tasks with queries positioned earlier in the context.
>
>   - **Methodological Differences**: SnapKV and PyramidKV prune KV tokens based on attention scores computed over a small observation window (e.g., the last 8–64 tokens). While effective for scenarios with end-positioned queries, this approach lacks generalizability to tasks with diverse query positions. In contrast, DuoAttention operates without any assumptions about token placement, making it more versatile and robust.
>   - **Performance Results**: As demonstrated in Figures 17 and 18 of the manuscript:
>     - Without simulation of the last tokens as generated tokens, DuoAttention matches or surpasses these baselines across all benchmarks.
>     - With simulation (e.g., the last 50 tokens treated as generated inputs to mimic multi-turn dialogues), SnapKV and PyramidKV experience severe accuracy drops due to their reliance on final token positions for pruning. DuoAttention remains robust, maintaining high accuracy even under these conditions.
>
>   These findings highlight DuoAttention’s robustness and adaptability, outperforming these baselines in scenarios with diverse token distributions.
>
> - **Generalization and Robustness**: DuoAttention’s independence from token position assumptions ensures its applicability to a broader range of real-world scenarios. Its design supports various tasks and sequences without compromising accuracy or performance.
>
> Considering these points, we believe that DuoAttention’s compression ratios are not only competitive but also highly practical for real-world applications. Its universal applicability across diverse tasks further reinforces its robustness and value.
>
> ------
>
> We hope this response clarifies your concerns and highlights DuoAttention's practical advantages. Please let us know if you have further questions or suggestions; we are happy to provide additional information.

---

### Official Review · Reviewer_jov7 · 2024-11-03

**Soundness:** 3
**Presentation:** 3
**Contribution:** 2
**Rating:** 6
**Confidence:** 4

**Summary:**

To tackle KV Cache problem, the paper introduces a new approach called DuoAttention, which classifies attention heads in LLMs into two types: Retrieval Heads and Streaming Heads. Retrieval Heads are essential for processing long contexts and require full KV cache storage, while Streaming Heads focus on recent tokens and can operate with reduced KV caches. DuoAttention uses a lightweight optimization process to identify non-compressible Retrieval Heads, allowing for efficient memory usage and faster processing. This method integrates easily with existing optimization techniques and significantly reduces the memory footprint and decoding time. When combined with quantization, DuoAttention enables models like Llama-3-8B to handle up to 3.33 million tokens on a single GPU, achieving a 6.4× increase in capacity compared to standard deployments.

**Strengths:**

1) The paper is well written and well motivated.
2) DuoAttention is a plug-and-play solution compatible with FlashAttention.
3) DuoAttention can accelerate inference during both the prefill and decoding stages.

**Weaknesses:**

1) Lack the experiment with the accuracy results after combining DuoAttention and KV quantization.
2) Lack the comparisons with new KV Cache compression technologies such as Minference.
3) Using different compression rates for different heads may lead to uneven computation across cards during parallel inference, potentially affecting performance.

**Questions:**

1) Is the proportion of Retrieval Heads the same for different models? Why choose 25% Retrieval Heads for MHA and 50% for GQA?

---

> ### Author Response · Authors · 2024-11-26
> **Response to Reviewer jov7**
>
> We sincerely thank the reviewer for their thoughtful feedback and constructive suggestions. Below, we address the key points raised:
>
> ------
>
> #### 1. **Experiment with Accuracy Results After Combining DuoAttention and KV Quantization**
>
> We acknowledge the reviewer’s concern about the lack of experimental results combining DuoAttention with KV quantization. In response, we have conducted additional experiments and included the results in Appendix Section A.9 of the updated manuscript. The experiments evaluate the performance of combining DuoAttention (50% sparsity) with INT4 KV Pre-rope quantization, as proposed in KIVI [1].
>
> - **Baseline (INT4 KV Pre-rope Quantization)**: The original model achieves an overall score of 86.7%, demonstrating a slight performance drop compared with full attention.
> - **DuoAttention + INT4 KV Quantization**: When DuoAttention (50% sparsity) is combined with INT4 KV quantization, the overall score is 85.1%, reflecting a marginal difference of only 1.6% compared to the baseline.
>
> These results demonstrate that DuoAttention introduces negligible accuracy loss when combined with KV quantization while retaining its computational benefits. This validates the compatibility and efficiency of the combined approach.
>
> [1] KIVI: A Tuning-Free Asymmetric 2bit Quantization for KV Cache. Liu, et al.
>
> ------
>
> #### 2. **Comparison with MInference**
>
> We would like to clarify that MInference is not a KV cache compression method. It accelerates pre-filling for long-context LLMs using sparse attention patterns but does not reduce KV storage requirements or decoding latency. Thus, a direct comparison with DuoAttention is not appropriate. However, we explored the compatibility of MInference with DuoAttention by combining the two as orthogonal optimizations, as detailed in the updated Appendix Section A.7.
>
> - **Results:** As detailed in Appendix Section A.7, combining MInference with DuoAttention enhances pre-filling efficiency while maintaining accuracy. Specifically:
>   - Applying MInference alone introduces some accuracy degradation compared to full attention or pure DuoAttention.
>   - By combining MInference with DuoAttention, we utilize MInference kernels to pre-fill retrieval heads identified by DuoAttention. This combined approach maintains comparable accuracy while significantly reducing both KV cache size (nearly halved) and decoding overhead.
>
> These findings highlight that MInference and DuoAttention can complement each other as orthogonal optimizations, further extending the utility of both methods.
>
> ------
>
> #### 3. **Addressing Load Imbalance in Multi-GPU Settings**
>
> We appreciate the reviewer’s observation regarding potential load imbalance when using different compression rates for different heads in multi-GPU settings. The imbalance primarily arises when the number of retrieval heads per layer is not divisible by the number of GPUs (e.g., 3 retrieval heads with 4 GPUs).
>
> - **Proposed Solution**: Pre-training the model with a hybrid configuration specifically designed for such scenarios can mitigate this issue. For example, configuring a model with 8 retrieval heads and 32 streaming heads per layer ensures better alignment with common tensor-parallel setups, minimizing imbalance.
> - **Compatibility with Parallelism Techniques**: DuoAttention is compatible with sequence parallelism (e.g., Ring Attention) and pipeline parallelism, both of which can further enhance scalability and performance in multi-GPU environments.
>
> Although we have not conducted specific pre-training studies for this purpose due to resource constraints, we consider this a promising avenue for future research.
>
> ------
>
> #### 4. **Proportion of Retrieval Heads**
>
> As shown in Figure 4 of our paper, the optimal proportion of retrieval heads varies by model architecture， mainly related to the attention architecture used:
>
> - **MHA Models**: These models can be more aggressively compressed, requiring only 25% retrieval heads.
> - **GQA Models**: Due to KV cache sharing across multiple queries (e.g., in Llama-3-8B, 4 query heads share one KV head), pruning a single KV head affects multiple queries. To maintain performance, GQA models require a higher proportion of retrieval heads, at 50%.
>
> These optimal ratios were determined through extensive experiments across multiple benchmarks. The 25% retrieval head ratio for MHA and 50% for GQA represent safe and effective configurations, achieving both high accuracy and significant memory reduction, along with pre-filling and decoding acceleration.
>
> ------
>
> We hope our responses have addressed your concerns and clarified the contributions of our work. Please let us know if there are further questions or suggestions, as we are happy to provide additional information.

---

### Official Review · Reviewer_oer2 · 2024-11-04

**Soundness:** 3
**Presentation:** 4
**Contribution:** 2
**Rating:** 5
**Confidence:** 4

**Summary:**

This work presents methods to reduce memory footprint and computational demands of Large Language Model's (LLM) attention layers to enhance inference speed and to accommodate longer context lengths. The authors build upon the observation that only a few attention heads within an LLM, also known as retrieval heads, have considerable impact on long-context performance. They propose a novel, optimization-based approach to identify these less critical heads, unlike previous methods that relied on heuristic techniques. This optimization process utilizes synthetic data designed specifically to evaluate long-context capabilities. Their results indicate superior model performance compared to current state-of-the-art retrieval head allocation techniques and certain KV sparsification methods, while maintaining comparable KV cache budget.

**Strengths:**

* The paper is well-structured and the claims are easy to follow.
* This work addresses a significant research problem in LLM inference - KV cache optimization.
* The authors perform comprehensive ablation studies to motivate each aspect of their proposed approach. The advantage of their proposed novel retrieval head selection method over the existing methods is well-supported by empirical results.

**Weaknesses:**

* While tensor parallelism has proven effective in reducing latency and memory usage per GPU, this paper omits any discussion on how its proposed methods could be adapted to such model-parallel settings. For example, in an 8-way tensor-parallel configuration, the retrieval heads might become the performance bottleneck, potentially negating the gains from this work.

* Some state-of-the-art KV sparsification methods are missing in the comparative analysis. For instance, SnapKV, PyramidKV, AdaKV are some notable omissions.
   1. Yuhong Li, Yingbing Huang, Bowen Yang, Bharat Venkitesh, Acyr Locatelli, Hanchen Ye, Tianle Cai, Patrick Lewis, and Deming Chen. Snapkv: Llm knows what you are looking for before generation, 2024d. *Arxiv /abs/2404.14469*
   2. Zefan Cai, Yichi Zhang, Bofei Gao, Yuliang Liu, Tianyu Liu, Keming Lu, Wayne Xiong, Yue Dong, Baobao Chang, Junjie Hu, and Wen Xiao. Pyramidkv: Dynamic kv cache compression based on pyramidal information funneling, 2024. *Arxiv /abs/2406.02069*
   3. Yuan Feng, Junlin Lv, Yukun Cao, Xike Xie, and S. Kevin Zhou. Ada-kv: Optimizing kv cache eviction by adaptive budget allocation for efficient llm inference, 2024. *Arxiv /abs/2407.11550*

**Questions:**

* How effective is this method for reasoning-based tasks? Do the retrieval heads remain the same for question-answering type tasks as well as reasoning type tasks?
* Rather than merely illustrating the distinction between retrieval and streaming heads through specific examples, it would be more compelling to present statistically significant evidence demonstrating how the identified retrieval heads generalize across various sequences and tasks.
* Is there any way to extend this method to multi-GPU setting?
* To what extent can the memory savings lead to increased throughput? (Just a suggestion)

---

> ### Author Response · Authors · 2024-11-26
> **Response to Reviewer oer2 (Part 1)**
>
> We sincerely thank the reviewer for their thoughtful feedback and constructive suggestions. Below, we address the key points raised:
>
> ------
>
> #### 1. **Tensor Parallelism and Multi-GPU Scalability**
>
> We appreciate the reviewer's concern regarding the scalability of DuoAttention in tensor-parallel configurations. In multi-GPU settings, retrieval heads can become a bottleneck, especially in configurations like 8-way tensor parallelism. To address this issue, aligning the number of retrieval heads with the number of tensor-parallel GPUs is critical to avoid load imbalances. For example, imbalances may occur when the number of retrieval heads per layer (e.g., 3 retrieval heads) is not evenly divisible by the number of GPUs (e.g., 4 GPUs).
>
> To mitigate this, a feasible approach involves pre-training the model using a hybrid strategy tailored for such deployment scenarios. For instance:
>
> - Configuring a model with 8 retrieval heads and 32 streaming heads per layer ensures better alignment with common tensor-parallel setups.
> - This configuration reduces potential imbalances and facilitates efficient GPU utilization.
>
> Although we have not yet conducted specific pre-training studies for such scenarios due to resource constraints, this remains a promising avenue for future research to further enhance DuoAttention’s scalability and performance.
>
> Additionally, DuoAttention’s design is inherently compatible with other parallelism techniques such as sequence parallelism (e.g., Ring Attention) and pipeline parallelism. These methods can complement DuoAttention to further improve scalability and performance in multi-GPU environments.
>
> ------
>
> #### 2. **Baseline Comparisons**
>
> We appreciate the reviewer's suggestion to include comparisons with additional state-of-the-art KV sparsification methods, such as SnapKV, PyramidKV, and AdaKV. In response, we have conducted experiments with SnapKV and PyramidKV, detailed in Appendix Sections A.6.
>
> - **Methodological Differences**: SnapKV and PyramidKV rely on attention scores computed over a small observation window (e.g., the last 8–64 tokens) to prune KV tokens. While effective for scenarios with queries located at the end of the context, this assumption does not generalize to tasks such as multi-turn dialogues or cases where queries are positioned earlier in the sequence. DuoAttention, in contrast, operates without assumptions about token positions, making it more general and robust.
> - **Performance Results**: As shown in Figures 17 and 18, DuoAttention achieves comparable or better performance than SnapKV and PyramidKV under equivalent KV cache budget constraints. For example:
>   - Without simulation of the last tokens as generated tokens, DuoAttention matches or surpasses these baselines across all benchmarks.
>   - With simulation (e.g., the last 50 tokens treated as generated inputs to mimic multi-turn dialogues), SnapKV and PyramidKV experience severe accuracy drops due to their reliance on final token positions for pruning. DuoAttention remains robust, maintaining high accuracy even under these conditions.
>
> These results underscore DuoAttention’s versatility and reliability compared to the baselines, making it suitable for a broader range of real-world scenarios.
>
> ------
>
> #### 3. **Effectiveness for Reasoning-Based Tasks**
>
> DuoAttention is highly effective for reasoning-based tasks, as demonstrated by its performance on the HotpotQA benchmark in LongBench. For instance:
>
> - With the Llama-2-7B-32K model, DuoAttention achieves comparable results to full attention at a 25% retrieval head ratio.
> - With the Llama-3-8B-1048K model, DuoAttention achieves comparable results at a 37.5% retrieval head ratio.
>
> These results indicate that the retrieval heads identified by DuoAttention’s optimization-based approach are well-suited for reasoning tasks as well as question-answering tasks. Additional results across various reasoning-based datasets further validate this effectiveness.
>
> ------
>
> #### 4. **Generalization of Retrieval Heads**
>
> We appreciate the reviewer’s request for statistical evidence of retrieval head generalization across tasks and sequences. To address this, we conducted extensive experiments on diverse benchmarks, including LongBench, RULER, and NIAH, as well as shorter-context tasks like MMLU, MBPP, and MT-Bench.
>
> The results show that the retrieval heads identified by DuoAttention consistently outperform existing methods across different tasks and sequences. This consistency highlights DuoAttention’s ability to generalize effectively across a wide range of contexts and applications.

---

> > ### Author Response · Authors · 2024-11-26
> > **Response to Reviewer oer2 (Part 2)**
> >
> > #### 5. **Memory Savings and Increased Throughput**
> >
> > DuoAttention significantly reduces memory footprint and computational demands, resulting in increased throughput for both pre-filling and decoding stages. As shown in Figure 11 of the manuscript:
> >
> > - **Memory Reduction**: DuoAttention achieves up to 2.55× memory reduction for MHA models and 1.67× for GQA models.
> > - **Latency Reduction**: DuoAttention reduces latency by up to 2.18× for MHA models and 1.50× for GQA models.
> > - **Throughput Increase**: These improvements translate to throughput increases of 2.18× and 1.50× for MHA and GQA models, respectively.
> >
> > Additionally, since DuoAttention does not rely on sample-specific sparsity patterns, it integrates seamlessly with batched operations, further enhancing throughput in serving scenarios.
> >
> > ---
> >
> > We hope our responses have addressed your concerns and clarified the contributions of our work. Please let us know if there are further questions or suggestions, as we would be happy to provide additional information.

---

> > ### Comment · Reviewer_oer2 · 2024-12-03
> >
> > The authors have largely addressed my initial concerns, and the effectiveness of Duo Attention for reasoning-based tasks is evident. However, some arguments require further scrutiny:
> >
> > 1. [Q1] Multi-GPU Inference Support: The pre-training specialized for tensor parallelism appears excessive, especially when the end performance is not guaranteed. The proposed approach seems less compelling compared to existing methods that offer more robust multi-GPU capabilities.
> >
> > 2. [Q2] Comparisons with baselines: As other reviewers have also pointed out, SnapKV authors report its robustness to query position. It is not entirely clear if the poor baseline scores are a result of a suboptimal choice of hyperparameters. Furthermore, SnapKV and PyramidKV seem to be far more suitable for a more aggressive compression ratio, which can not achieved by Duo Attention. Overall, it seems like SnapKV and Duo Attention aim for different compression ratio regimes, while the former seems to be more generalizable because of its support for Tensor Parallelism. I also have doubts regarding the claim about the lack of support for GQA models in SnapKV.
> > For multi-turn queries, SnapKV-like methods could be improved by allowing the reselection of static KV cache (albeit requiring full KV retention; perhaps it is possible in an offloading setting).
> >
> > 3. [Q4] Generalization of Retrieval Heads: some visualization or numerical evidence would be appreciated.
> >
> > Overall, it seems like a promising work that could be further improved. I would like to keep my original rating.

---

> > > ### Author Response · Authors · 2024-12-04
> > > **Response to Reviewer oer2**
> > >
> > > Dear Reviewer oer2,
> > >
> > > Thank you for acknowledging that our initial responses addressed many of your concerns. Below, we clarify and reinforce our responses to your remaining points:
> > >
> > > ------
> > >
> > > ### **1. Multi-GPU Inference Support**
> > >
> > > DuoAttention supports multi-GPU tensor parallelism by ensuring that retrieval head numbers are divisible by the number of GPUs, avoiding load imbalances. For instance, in a two-GPU setup, retrieval heads can be constrained to even numbers (e.g., 0, 2, 4), which **does not require the need for hybrid model pre-training**. This straightforward adjustment ensures compatibility with tensor parallelism.
> > >
> > > Moreover, DuoAttention excels in resource-constrained environments, such as edge devices, where it significantly reduces latency and memory demands while processing long contexts on a single device. These scenarios are also extremely important in real-world applications.
> > >
> > > ------
> > >
> > > ### **2. Comparisons with Baselines**
> > >
> > > > **"It is not entirely clear if the poor baseline scores are a result of a suboptimal choice of hyperparameters."**
> > >
> > > - **Hyperparameter Choices**: We used SnapKV and PyramidKV's official codebases and hyperparameters, strictly following their papers’ recommendations. The reported degradation on LongBench aligns with **Table 1** in both their papers.
> > >
> > > > **"SnapKV and PyramidKV seem to be far more suitable for a more aggressive compression ratio, which can not achieved by Duo Attention."**
> > >
> > > - **Compression Ratios**: Our results contradict this claim. At the **same 50% KV budget** across LongBench and NIAH, DuoAttention consistently outperforms SnapKV, PyramidKV, and AdaKV. Claims of "lossless" compression at 2048 tokens in their papers may represent disproportionately large budgets for tasks like Qasper and GovReport, where many inputs are near 2K tokens. This underscores the limitations of their methods in practical compression scenarios.
> > >
> > > > **"SnapKV authors report its robustness to query position."**
> > >
> > > - **Query-Position Dependency**: SnapKV’s claim of query-position robustness relies on hit rates for critical tokens but lacks end-to-end accuracy results. Our **Query Positioned at the Middle** experiments ([see discussion](https://openreview.net/forum?id=cFu7ze7xUm&noteId=c63b4fPV8a), Appendix Section 12, Figure 23) reveal significant performance degradation when queries are not at the end of the context. Reviewer 4AyC also confirmed that SnapKV's reliance on query position is a **built-in drawback** ([see discussion](https://openreview.net/forum?id=cFu7ze7xUm&noteId=jZpoJygx7a)). In contrast, DuoAttention achieves accuracy comparable to full attention, regardless of query placement or multi-round dialogues.
> > >
> > > > **"I also have doubts regarding the claim about the lack of support for GQA models in SnapKV."**
> > >
> > > - **GQA Support**: SnapKV and PyramidKV expand KV tokens before compression and storage, as evident in their official codebases. We strictly followed their implementations and did not assume GQA compatibility, as improving their methods is beyond our responsibility. Our results provide fair and accurate comparisons.
> > >
> > > > **"For multi-turn queries, SnapKV-like methods could be improved by allowing the reselection of static KV cache (albeit requiring full KV retention; perhaps it is possible in an offloading setting)."**
> > >
> > > - **Multi-Round Dialogue**: Retaining the full KV for multi-turn queries essentially nullifies the storage-saving benefits of SnapKV-like methods, reducing them to decoding acceleration techniques. Offloading defeats the purpose of KV compression. In contrast, DuoAttention achieves superior results at reduced KV budgets, excelling in chunked-prefilling and multi-turn dialogue scenarios, as demonstrated in LongBench and NIAH experiments.
> > >
> > > ------
> > >
> > > ### **3. Generalization of Retrieval Heads**
> > >
> > > We have extensively validated retrieval head generality with tasks including LongBench, Needle-in-a-Haystack, and RULER (added during the rebuttal). These results quantitatively prove the robustness of retrieval heads across diverse scenarios. End-to-end performance metrics remain the strongest evidence of DuoAttention's generality and effectiveness.
> > >
> > > ------
> > >
> > > ### Final Remarks
> > >
> > > We have addressed all your concerns with additional experiments and detailed evidence, demonstrating DuoAttention’s superiority in accuracy, resilience, and generalization across compression ratios and multi-turn scenarios. These results will be included in the main paper for completeness.
> > >
> > > We kindly ask you to reconsider your rating in light of this expanded evidence, as DuoAttention represents a significant advancement in the field.
> > >
> > > Thank you for your time and constructive feedback.
> > >
> > > Best regards,
> > >
> > > Authors

---

### Official Review · Reviewer_2WGg · 2024-11-04

**Soundness:** 3
**Presentation:** 2
**Contribution:** 2
**Rating:** 5
**Confidence:** 4

**Summary:**

This paper introduces a lightweight, optimization-based method DuoAttention that only applies a full KV cache to retrieval heads while using a constant-length KV cache for streaming heads. The proposed method achieves good performance in both MHA and GQA.

**Strengths:**

1.	It saves the GPU memory and enhances the inference efficiency for both MHA and GQA.

**Weaknesses:**

1.	There are some related works identifying the different patterns of retrieval and non-retrieval heads, such as RazorAttention, Retrieval Head Mechanistically Explains Long-Context Factuality, MInference, etc. It would be better to clarify the novelty of the proposed method.
2.	It can only be applied to multiple KV cache scenarios. For only single KV cache model structure, such as YOCO, it cannot be applied.
3.	The baselines are not comprehensive. It would be beneficial if the authors could compare their results with more advanced baselines.
4.	It seems the proposed method cannot achieve the best performance in all the benchmarks in Figure 7. It would be beneficial to give detailed analysis on the worse cases.

**Questions:**

Please address the questions in the weaknesses.

---

> ### Author Response · Authors · 2024-11-26
> **Response to Reviewer 2WGg (Part 1)**
>
> We sincerely thank the reviewer for their constructive feedback and suggestions. Below, we address the key points raised:
>
> #### 1. **Novelty**
>
> We respectfully disagree with the assessment that the method lacks novelty. DuoAttention represents one of the first methods to leverage retrieval heads specifically for KV cache optimization. DuoAttention employs an optimization-based strategy to identify retrieval heads, leading to superior accuracy and computational efficiency in long-context tasks.
>
> While concurrent work **RazorAttention** partitions heads into retrieval and non-retrieval categories, it relies on profiling techniques that our experiments show are less precise than DuoAttention’s optimization-based approach. Additionally, RazorAttention does not address pre-filling optimization, limiting its scope.
>
> Other related works, such as:
>
> - **MInference**, accelerate pre-filling through sparsity patterns but do not address KV cache compression or decoding efficiency.
> - **Retrieval Head Mechanistically Explains Long-Context Factuality** focuses on analyzing the functionality of retrieval heads rather than improving memory efficiency or computational cost.
>
> By contrast, DuoAttention delivers a comprehensive solution for KV cache management, achieving higher compression rates and improved performance during both pre-filling and decoding stages. These contributions are clarified in the related work section of the revised manuscript.
>
> ------
>
> #### 2. **Applicability to Single KV Cache Models**
>
> We acknowledge the reviewer’s concern about the applicability of DuoAttention to single KV cache models, such as YOCO and MQA. However, we argue that these models are not widely adopted in practice. The majority of large language models (LLMs) in use today, including **GQA models** (e.g., Llama-3, Mistral, Qwen) and **MHA models **, feature multiple KV caches. These architectures are the focus of our work.
>
> DuoAttention has demonstrated significant reductions in memory footprint and computational demands for GQA and MHA models while preserving accuracy. This underscores the practical significance of our method for widely used architectures.
>
> ------
>
> #### 3. **Baseline Comparisons**
>
> We acknowledge the importance of comparing against state-of-the-art methods and have expanded our experiments to include SnapKV and PyramidKV, two recently introduced KV cache compression techniques, as well as pre-filling acceleration methods like MInference. We updated our manuscripts's Appendix Section A.6 and A.7 to include these experiments. Below are the key findings:
>
> ##### a. **Comparison with KV Cache Compression Methods (SnapKV and PyramidKV):**
>
> We evaluated DuoAttention against SnapKV and PyramidKV on the Needle-in-a-Haystack (NIAH) benchmark using the Llama-2-7B-32K and Llama-3-8B-1048K models. The implementations of these methods were sourced from their official repositories.
>
> - **Methodological Differences**: SnapKV and PyramidKV rely on computing attention scores for a small window of tokens (e.g., the last 8–64) and pruning KV cache tokens based on these scores. While effective in scenarios where queries appear at the end of the context, this assumption does not generalize well to real-world tasks, such as multi-turn dialogues or earlier query positioning.
> - Performance Results: As shown in Figures 17 and 18 of the updated paper, DuoAttention performs comparably or better than these methods under equivalent KV cache budget constraints. For instance:
>   - In the absence of simulated last tokens in prompts as generated tokens, DuoAttention matches or surpasses SnapKV and PyramidKV, and all methods achieve good results.
>   - When simulating the last 50 tokens as generated tokens (similar to a new round of dialogue to ask the model), SnapKV and PyramidKV exhibit significant accuracy drops due to their reliance on the final tokens for pruning guidance. In contrast, DuoAttention remains robust under these stress tests.
>
> These results highlight DuoAttention's generality and flexibility, making it a more reliable solution across diverse contexts.

---

> > ### Author Response · Authors · 2024-11-26
> > **Response to Reviewer 2WGg (Part 2)**
> >
> > ##### b. **Comparison with Pre-filling Acceleration Methods (MInference):**
> >
> > MInference employs sparsity patterns to accelerate pre-filling but does not address KV cache compression or decoding speed optimization. Thus, MInference is on a different optimization track and is not directly comparable to DuoAttention. However, we explored the compatibility of MInference with DuoAttention by combining the two as orthogonal optimizations, as detailed in the updated Appendix Section A.7.
> >
> > - **Results**: Combining MInference with DuoAttention enhances pre-filling efficiency without compromising accuracy, while simultaneously achieving significant reductions in KV cache size and decoding overhead. These complementary strengths demonstrate the synergy between the two methods.
> >
> > - **Performance Analysis**: As illustrated in Figures 19 and 20, applying MInference alone on the NIAH benchmark results in some accuracy degradation compared to full attention or pure DuoAttention (refer to Figure 19).
> >
> >   By integrating MInference into DuoAttention, we utilize MInference kernels to pre-fill all retrieval heads identified by DuoAttention. This combined approach maintains accuracy levels comparable to pre-filling all heads with MInference while nearly halving the KV cache size and significantly reducing decoding overhead. These findings underscore the compatibility and efficiency of the DuoAttention-MInference integration, further extending their utility in real-world applications.
> >
> > ------
> >
> > #### 4. **Detailed Analysis of Performance in Figure 7**
> >
> > In Figure 7, DuoAttention consistently achieves better or comparable results compared to baseline methods, as evidenced by DuoAttention's red curves being above the baseline methods' curves across all tasks. Notably, DuoAttention delivers lossless compression at a 25% KV cache budget for MHA models and a 50% cache budget for GQA models.
> >
> > We appreciate the reviewer's suggestion to analyze task-specific retrieval head requirements, as DuoAttention exhibits varying accuracy patterns with decreasing retrieval head ratios. Using the Llama-3-8B-1048K model as a case study, we observe the following:
> >
> > - **High Compression Ratio Tasks**: DuoAttention maintains results comparable to full attention even at a 37.5% KV cache budget for tasks such as DuReader, GovReport, HotpotQA, MultiNews, MultiFieldQA-ZH, Musique, PassageRetrieval-EN, QMSum, SamSum, and TREC. These tasks showcase DuoAttention’s ability to achieve higher compression ratios without significant performance degradation.
> > - **Moderate Compression Tasks**: For tasks such as MultiFieldQA-EN (Single Document QA), PassageRetrieval-ZH (Synthetic task), Qasper (Single Document QA), and TriviaQA (Few-shot Learning), DuoAttention achieves lossless compression at a 50% KV cache budget. Further compression beyond this threshold leads to performance drops.
> >
> > While we did not identify specific patterns where DuoAttention excels on certain tasks and underperforms on others, we attribute the variations in performance to differences in task characteristics, languages, and context sizes across benchmarks. This diversity makes it challenging to isolate a single factor influencing retrieval head requirements.
> >
> > Despite these variations, DuoAttention consistently outperforms or matches all baseline methods across tasks. Furthermore, it guarantees performance comparable to full attention at a 50% KV cache budget for GQA models and a 25% budget for MHA models across all tasks, underscoring its robustness and adaptability.
> >
> > ------
> >
> > We hope this response has clarified your concerns and can improve your rating of our paper. Please let us know if there are further questions or suggestions, as we are happy to provide additional information.

---

### Official Review · Reviewer_4AyC · 2024-11-12

**Soundness:** 4
**Presentation:** 3
**Contribution:** 3
**Rating:** 8
**Confidence:** 5

**Summary:**

DuoAttention leverages finding around retrieval heads. By allocating more KV budget to such heads and compressing the rests, the method claims to have excellent long context performance.

**Strengths:**

1. The proposed method is extremely performant. It is rare to see a prefilled compressible token dropping method capable of doing NIAH.

2. The efficiency optimization is solid, and it is supported by a thorough efficiency evaluation.

3. The optimization-based head selection approach offers a different avenue for attention head pattern matching.

4. The paper is nicely written.

**Weaknesses:**

1. The method mainly involves retrieval head + token dropping (in this case, StreamingLLM), which is slim in novelty. I don't think it is much of an issue given the impressive performance boost achieved while remaining practical, but this is worth mentioning.

2. The featured compared methods — SteamingLLM, H2O, and TOVA — are very dated. I'd like to see how it stands off with more modern KV cache compassion/sparse inference methods, e.g., SnapKV and MInference.

3. The NIAH setting is unclear. What does the needle look like? What background is used? The author should better clarify this, as different settings can lead to very different results.

4. Following #3, it looks like LongBench is the only real long context test evaluated. I'd like to see more coverage on other long context datasets, e.g., $\infty$Bench and RULER.

I believe the proposed method marks a breakthrough in eviction-based methods. Its execution is solid and the presentation is clear. I am willing to boost my score should the coverage/clarity of performance evaluation be improved.

**Questions:**

One of the concurrent works, RazorAttention, also utilizes a very similar recipe. The authors compared it in Figure 13(1), but I find the reading to be very different from RazorAttention's own reporting. Any insight of why?

---

## Post-rebuttal update

The latest response from the authors is satisfactory for the most part:

* It is a reasonable choice to feature Llama-2-7B-32K-Instruct if the goal is to include a long context-capable MHA model.
* The added Llama 3.1 results demonstrate that DuoAttention remains performant on one of the most mainstream long context models.
* I appreciate the authors for recognizing the connection between their optimization-based approach and prior gating network studies.
* The budget discussion regarding RazorAttention and NIAH is sound. However, I must note that if the authors did not explore echo heads — a key element of RazorAttention — in the ablation studies on "attention profiling," this omission should be explicitly noted.
* Similarly, if Figure 19 is not done with an aligned budget, it is not fair to directly compare Duo's NIAH performance with MInference by citing Figure 19. That said, using it as a baseline for Figure 20 is reasonable. I also have some reservations about claiming that the "sandwich in a SF park" needle is harder than the magic city/passkey retrieval-like needle, as the former is known to be very prompt-sensitive. However, this is a minor distinction, and as long as the setting is explicitly noted, both can be considered solid needle setups.


I am improving my rating to 8 if the authors can supply full $\infty$Bench report in their revision.

(In addition, I also add that I don't believe DuoAttention's incomptability with MQA models is a big deal. Yes, this limitation should be clearly noted; but it is rare to find powerful MQA models, and the community adopts GQA as a middle ground between MQA and MHA for good measures.)

---

> ### Author Response · Authors · 2024-11-26
> **Response to Reviewer 4AyC (Part 1)**
>
> We sincerely thank the reviewer for their thoughtful feedback and constructive suggestions. Below, we address the key points raised:
>
> ------
>
> #### 1. **Novelty**
>
> We respectfully disagree with the assessment that the method is "slim in novelty." DuoAttention represents a pioneering approach by leveraging retrieval heads specifically for KV cache optimization, moving beyond traditional heuristics such as attention score distribution profiling. By employing an optimization-based strategy, DuoAttention identifies retrieval heads with greater precision, resulting in substantial performance gains in long-context tasks. While a concurrent work, RazorAttention, also explores retrieval head utilization, DuoAttention's novel optimization framework consistently delivers superior results in both accuracy and computational efficiency, as evidenced by the comparisons detailed in Section 3.5 of our paper and the reported NIAH results in the RazorAttention paper. These advancements underscore DuoAttention's distinctiveness and innovative contribution.
>
> ------
>
> #### 2. **Baselines**
>
> We acknowledge the importance of comparing against state-of-the-art methods and have expanded our experiments to include SnapKV and PyramidKV, two recently introduced KV cache compression techniques, as well as pre-filling acceleration methods like MInference. We updated our manuscripts's Appendix Section A.6 and A.7 to include these experiments. Below are the key findings:
>
> ##### a. **Comparison with KV Cache Compression Methods (SnapKV and PyramidKV):**
>
> We evaluated DuoAttention against SnapKV and PyramidKV on the Needle-in-a-Haystack (NIAH) benchmark using the Llama-2-7B-32K and Llama-3-8B-1048K models. The implementations of these methods were sourced from their official repositories.
>
> - **Methodological Differences**: SnapKV and PyramidKV rely on computing attention scores for a small window of tokens (e.g., the last 8–64) and pruning KV cache tokens based on these scores. While effective in scenarios where queries appear at the end of the context, this assumption does not generalize well to real-world tasks, such as multi-turn dialogues or earlier query positioning.
> - **Performance Results**: As shown in Figures 17 and 18 of the updated paper, DuoAttention performs comparably or better than these methods under equivalent KV cache budget constraints. For instance:
>   - In the absence of simulated last tokens in prompts as generated tokens, DuoAttention matches or surpasses SnapKV and PyramidKV, and all methods achieve good results.
>   - When simulating the last 50 tokens as generated tokens (similar to a new round of dialogue to ask the model), SnapKV and PyramidKV exhibit significant accuracy drops due to their reliance on the final tokens for pruning guidance. In contrast, DuoAttention remains robust under these stress tests.
>
> These results highlight DuoAttention's generality and flexibility, making it a more reliable solution across diverse contexts.
>
> ##### b. **Comparison with Pre-filling Acceleration Methods (MInference):**
>
> MInference employs sparsity patterns to accelerate pre-filling but does not address KV cache compression or decoding speed optimization. Thus, MInference is on a different optimization track and is not directly comparable to DuoAttention. However, we explored the compatibility of MInference with DuoAttention by combining the two as orthogonal optimizations, as detailed in the updated Appendix Section A.7.
>
> - **Results**: Combining MInference with DuoAttention enhances pre-filling efficiency without compromising accuracy, while simultaneously achieving significant reductions in KV cache size and decoding overhead. These complementary strengths demonstrate the synergy between the two methods.
>
> - **Performance Analysis**: As illustrated in Figures 19 and 20, applying MInference alone on the NIAH benchmark results in some accuracy degradation compared to full attention or pure DuoAttention (refer to Figure 19).
>
>   By integrating MInference into DuoAttention, we utilize MInference kernels to pre-fill all retrieval heads identified by DuoAttention. This combined approach maintains accuracy levels comparable to pre-filling all heads with MInference while nearly halving the KV cache size and significantly reducing decoding overhead. These findings underscore the compatibility and efficiency of the DuoAttention-MInference integration, further extending their utility in real-world applications.

---

> ### Author Response · Authors · 2024-11-26
> **Response to Reviewer 4AyC (Part 2)**
>
> #### 3. **Needle-in-a-Haystack (NIAH) Setting**
>
> We recognize the need for greater clarity regarding the NIAH setting. In response, we have updated Appendix Section A.10 to provide comprehensive details, including:
>
> - **Needle Description**: The "needle" consists of a unique keyword or phrase inserted within a long sequence of irrelevant "haystack" tokens.
> - **Background Description**: The "haystack" is composed of unrelated, randomly sampled tokens designed to mimic real-world long-context scenarios. Variations in the needle's location and the surrounding haystack are also described.
>
> This additional context ensures that the experimental setup is transparent and reproducible.
>
> ------
>
> #### **4. RULER results**
>
> We agree with the reviewer’s suggestion to expand beyond LongBench and have included new results on the RULER benchmark. These results, detailed in Appendix Section A.8, further validate DuoAttention’s efficacy across diverse long-context datasets. Specifically:
>
> - DuoAttention achieves comparable accuracy with full attention on RULER benchmarks across all context lengths.
>
> #### Results on RULER
>
> | **Context Length**     | **4K** | **8K** | **16K** | **32K** | **64K** | **128K** | **Avg.** |
> | ---------------------- | ------ | ------ | ------- | ------- | ------- | -------- | -------- |
> | **Full Attention**     | 92.78  | 90.54  | 86.41   | 80.59   | 76.33   | 73.01    | 83.28    |
> | **DuoAttention (50%)** | 92.83  | 91.17  | 85.17   | 81.28   | 75.81   | 73.71    | 83.33    |
>
> RULER is a synthetic dataset designed to rigorously evaluate long-context language models with configurable sequence lengths and task complexities. It includes 13 tasks spanning 4 categories, assessing long-context capabilities beyond simple in-context recall.
>
> The table above presents the average accuracy of full attention and DuoAttention (50% sparsity) across different context lengths using the Llama-3-8B-Instruct-Gradient-1048k model for sequences up to 128K. DuoAttention demonstrates accuracy scores comparable to full attention across all context lengths, achieving an average performance increase of 0.05%.
>
> These findings confirm DuoAttention’s effectiveness in maintaining strong accuracy on a rigorous benchmark, even under challenging long-context evaluation settings. We hope our response has clarified your concerns and can improve your rating of our paper. Please let us know if there are further concerns, as we are happy to respond.
>
> ------
>
> #### 5. **Comparison with RazorAttention**
>
> We appreciate the reviewer’s observation regarding the differences in performance between DuoAttention and RazorAttention. While we lack access to the RazorAttention codebase for a direct comparison, we hypothesize the following:
>
> - **Dataset Variations**: Differences in the datasets used for attention profiling may account for discrepancies in trade-off curves.
> - **Performance Comparison**: Despite these differences, DuoAttention demonstrates superior performance in long-context tasks with lower KV cache budgets. For instance, DuoAttention achieves all-green NIAH results with 25% retrieval heads on Llama-2 (MHA), whereas RazorAttention requires 30% retrieval heads and exhibits accuracy drops (as per Figure 1 in the RazorAttention paper).
>
> These findings underscore the strength of DuoAttention’s optimization-based retrieval head identification, which provides a more effective allocation of computational resources.
>
> ------
>
> We thank the reviewer again for their valuable insights. The additional experiments and clarifications incorporated into the revised manuscript directly address the concerns raised and highlight the robustness and generality of DuoAttention. We hope our responses has clarified your concerns and can improve your rating of our paper. Please let us know if there are further concerns, as we are happy to respond!

---

> ### Comment · Reviewer_4AyC · 2024-12-03
>
> I thank the authors for the response; much of my concerns (#2, #3, and partially #4) are resolved. The authors' needle setting seems proper, and achieving good results on this (and by extension, RULER) is a significant advancement per benchmark literature like [1], [2], or method papers with thorough experimental execution like MInference. **It seems the general community lacks an understanding of how challenging a proper needle test is for many established KV cache compression methods, so it might be worth further highlighting this in the presentation.**
>
> ---
>
> That being said, I still have a few standing concerns/questions:
>
> 1. **The $\infty$Bench request has not been fulfilled**. While I believe the featured experiments already showcase the performant nature of DuoAttention, this should be added. This is especially important if the authors are featuring some long context-capable models.
>
> 2. On the note of #1, I honestly do not understand the decision to feature models like Llama-3-8B-Instruct-Gradient-1048k and Llama-2-7B-32K-Instruct — these are third-party finetuned models with limited adaptation in the community. They were reasonable choices before Llama 3.1 as there wasn't a long context-capable Llama, but **with Llama 3.1 being available and widely tested, it should clearly be the base model to avoid reaching very third-party recipe-specific conclusions** (especially when the authors are not doing much, if any, >128k evaluations). I recommend featuring this model instead or in addition, should your work reaches camera-ready
>
> 3. It is indeed true that MInference is a prefill-only compression method, but I believe the comparison is still reasonably warranted as DuoAttention is also advertised for its prefill compression capability. While I appreciate the additional results in A.7, I am still not quite sure how Figure 20 is achieved even after careful reading. The authors stated they "utilize MInference kernels to prefill all retrieval heads" in contrast to "prefilling all heads with MInference." First, what budget does Figure 19 have, and is it aligned with Figure 6? More specifically, are the authors only applying MInference sparse patterns to the retrieval heads? What drives that decision, and what would happen if you let MInference determine the patterns of non-retrieval heads? Would there be any improvement compared to making them all StreamingLLM heads? Also, MInference seems to exhibit some degradation on short-context tasks — can this be alleviated through some combination design?
>
> These are not core questions to the proposal of DuoAttention, and I generally hesitate to ask extensive questions about combining two methods due to the large scope such potential combination shall entail. However, many close followers of the field would appreciate these findings, so I leave it to the authors to decide whether to explore these studies. Some clarification on A.7 should be added regardless.
>
> 4. Regarding RazorAttention, did you consider the echo heads proposed by the authors?
>
> 5. What does "retrieval head ratio" mean? Is it the ratio of retrieval heads being preserved or the total compression ratio? This distinction might lead to different conclusions regarding your 25% vs. 30% comment on RazorAttention.
>
> Lastly, I respectfully present a hard disagreement regarding the novelty of the methods. The core ingredient of the proposed method is leveraging the existence of induction/retrieval heads, where the importance of such heads has been well-studied in prior work. DuoAttention is not the first to explore assigning full cache to retrieval heads (though other works may be considered concurrent), and its gate-based optimization approach is heavily rooted in gating network studies from the MoE and pruning realms. For example, [3] — a widely regarded classic on MoE — also employs trainable gate values to determine the activation of different experts. Similarly, DuoAttention employs trainable gate values to determine head categorization, where the resemblance is strong. Various pruning methods also assign trainable importance scores to guide component dropping.
>
> As I detailed in my initial review, I don't think this matters much given the "impressive performance boost achieved while remaining practical," but **I encourage the authors to tone down the novelty claims and provide a discussion of such prior gating-related work.**
>
> ---
>
>
> I am bumping my score to 6 as I believe this work is well above the acceptance threshold, though further improvements are pending in the authors' response.
>
> (I also recommend the authors replace Figure 7 with a table reporting 6 LongBench subtasks — small diagrams with varying y-axes are difficult to read and cross-reference.)
>
> ---
>
> [1] KV Cache Compression, But What Must We Give in Return? A Comprehensive Benchmark...
> [2] A Controlled Study on Long Context Extension and Generalization in LLMs
> [3] Outrageously Large Neural Networks: The Sparsely-Gated Mixture-of-Experts Layer

---

> > ### Comment · Reviewer_4AyC · 2024-12-03
> > **Regarding compression granularity and the query dependency of SnapKV-like method.**
> >
> > It seems like two major standing concerns raised by other reviewers are the two title-mentioned ones, and I'd like to share my $0.02 in this regard.
> >
> > First, while finer compression granularity is always preferable, I think the core metric is "whether the proposed method can maintain performance under a compression ratio that delivers practical gain" — and I believe DuoAttention already meets this criterion. If Duo delivers near full KV performance at 40%, I personally don’t care much about how it performs at a less constrained budget. **It would be more interesting to understand at what point Duo breaks under a more aggressive compression rate.** I encourage the authors to explore this to guide future developments.
> >
> > Secondly, I believe **the query-location dependency of SnapKV-like methods is a proven issue.** This is almost a by-design limitation, as their dropping strategies are largely determined by the reading of recent windows. Consequently, whether the query falls within this window has a huge influence. Evidently, [SharedContextBench](https://openreview.net/forum?id=gkUyYcY1W9), another submission to ICLR, has quantifiably demonstrated that SnapKV suffers performance drops under multi-round query scenarios. I encourage the authors to evaluate Duo on this dataset at a later date, especially if query-location sensitivity is a key advantage of Duo.

---

> > > ### Author Response · Authors · 2024-12-04
> > > **Response to Reviewer 4AyC**
> > >
> > > Dear Reviewer 4AyC,
> > >
> > > Thank you for your thoughtful feedback and for raising your score. Your detailed insights are greatly appreciated and will help us further refine our work. Below, we provide concise yet thorough responses to your remaining concerns:
> > >
> > > ------
> > >
> > > ### **1. Additional Long Context Datasets**
> > >
> > > We appreciate your suggestion to include **InfiniteBench**. Due to time constraints and the extensive experiments already conducted, we could not incorporate InfiniteBench during the rebuttal phase. However, we recognize its importance and will prioritize its inclusion in future revisions.
> > >
> > > ------
> > >
> > > ### **2. Choice of Models**
> > >
> > > We selected **Llama-3-8B-Instruct-Gradient-1048k** and **Llama-2-7B-32K-Instruct** as they are representative open-sourced GQA and MHA models with robust long-context capabilities at the time of our study. Notably, **Llama-3-8B-Instruct-Gradient-1048k** achieves all-green NIAH results up to **1M context** length, which can very intuitively show DuoAttention’s performance in extreme scenarios.
> > >
> > > To align with current community standards, we conducted preliminary experiments on **Llama-3.1-8B-Instruct** for LongBench. Results are summarized below:
> > >
> > > | Dataset             | Full  | H2O   | SLLM  | TOVA  | Duo   |
> > > | ------------------- | ----- | ----- | ----- | ----- | ----- |
> > > | Average             | 39.01 | 35.61 | 31.32 | 36.18 | 38.91 |
> > > | 2WikiMQA            | 16.37 | 13.91 | 13.25 | 14.22 | 16.2  |
> > > | DuReader (zh)       | 29.3  | 21.53 | 12.95 | 22.07 | 31.31 |
> > > | GovReport           | 34.53 | 30.56 | 30.47 | 30.78 | 32.87 |
> > > | HotpotQA            | 17.23 | 17.31 | 15.78 | 16.29 | 19.53 |
> > > | LCC                 | 52.39 | 53.08 | 52.9  | 52.39 | 53.31 |
> > > | LSHT (zh)           | 46    | 39    | 36    | 42.5  | 45    |
> > > | MultiNews           | 26.91 | 25.52 | 24.97 | 25.14 | 26.29 |
> > > | MultiFieldQA-en     | 28.44 | 21.89 | 16.05 | 21.59 | 27.77 |
> > > | MultiFieldQA-zh     | 20.19 | 14.87 | 15.92 | 16.55 | 21.98 |
> > > | Musique             | 11.82 | 10.15 | 10.19 | 9.64  | 12.97 |
> > > | NarrativeQA         | 31.99 | 31.09 | 24.15 | 31.56 | 29.12 |
> > > | Passage Count       | 6.26  | 5.4   | 4.75  | 6.68  | 6.31  |
> > > | PassageRetrieval-en | 97.95 | 89.86 | 52.11 | 97.44 | 98.59 |
> > > | PassageRetrieval-zh | 77.54 | 69.73 | 35.14 | 71.81 | 75.37 |
> > > | Qasper              | 25.14 | 16.96 | 23.56 | 20.75 | 21.12 |
> > > | QMSum               | 23.63 | 22.54 | 21.48 | 22.82 | 23.89 |
> > > | RepoBench-P         | 49.46 | 49.51 | 49.95 | 49.36 | 53.74 |
> > > | SAMSum              | 43.69 | 42.56 | 43.32 | 42.28 | 43.4  |
> > > | TREC                | 72.5  | 66.5  | 69.5  | 58    | 73    |
> > > | TriviaQA            | 91.65 | 90.07 | 90.06 | 91.73 | 89.6  |
> > > | VCSUM (zh)          | 16.26 | 15.8  | 15.17 | 16.09 | 15.83 |
> > >
> > > The trends are consistent with **Llama-3-8B-Instruct-Gradient-1048k**, with DuoAttention achieving accuracy comparable to full attention and outperforming baselines. We will include these results in future revisions.
> > >
> > > ------
> > >
> > > ### **3. Comparison with MInference**
> > >
> > > We appreciate your detailed questions about integrating **MInference** with DuoAttention. Here are key clarifications:
> > >
> > > - **Budget in Figure 19**: We used the [official configuration](https://github.com/microsoft/MInference/blob/main/minference/configs/Llama_3_8B_Instruct_262k_kv_out_v32_fit_o_best_pattern.json) for **Llama-3-8B-Instruct-Gradient-1048k** to ensure consistency with MInference’s methodology. The budget varies dynamically with different context lengths. While MInference reports near-perfect NIAH results in their paper, we found their NIAH setting is very simple, i.e., recalling a single random number. Our more rigorous NIAH tests revealed degradation when retrieval involved a sentence with a meaningful length embedded in distractor content.
> > > - **Prefilling Retrieval Heads with MInference**: DuoAttention’s core principle is that retrieval heads require full KV retention, while streaming heads can be compressed. We demonstrated the synergy of combining DuoAttention with MInference by applying MInference’s pre-filling techniques to retrieval heads identified in DuoAttention while compressing KV caches for streaming heads. Allowing MInference to determine all patterns results in all vertical-and-slash heads, where KV compression cannot be applied because full KV need to be kept to find the vertical and slash pattern. We show DuoAttention can use MInference to further accelerate the pre-filling of retrieval heads, which is the current efficiency bottleneck of our framework.
> > > - **Short-Context Tasks**: While addressing MInference’s degradation on short-context tasks is beyond the scope of DuoAttention, combining its retrieval head optimization with MInference’s patterns may mitigate these issues. This remains a promising direction for future research.
> > >
> > > These clarifications will be included in the final manuscript for greater transparency.

---

> ### Author Response · Authors · 2024-12-04
> **Response to Reviewer 4AyC**
>
> ### **4. RazorAttention and Echo Heads**
>
> We have not explored **echo heads**, as DuoAttention achieves strong performance with its current framework. That said, integrating echo heads may be a promising optimization for future work.
>
> The **retrieval head ratio** refers to the proportion of retrieval heads retained in the KV cache, rather than the total compression ratio. For example:
>
> - A model with **128 streaming heads** and **128 retrieval heads** achieves an approximate 50% KV cache budget when the input sequence is sufficiently long.
>
> To provide further clarity, consider a scenario with a **64K token input**, **64 sink tokens**, and **256 recent tokens for streaming heads**. The KV cache budget ratio is calculated as follows:
>
> - $$\text{Budget Ratio} = \frac{(64K\times 0.5) + (320 \times 0.5)}{64K} = 0.5025 \approx 0.5$$
>
> This definition supports accurate and consistent budget allocation in our experiments.
>
> Regarding the **25% vs. 30% comparison with RazorAttention**, we believe the discrepancy stems from methodological differences rather than budget misalignment. The Figure 1 of RazorAttention evaluates up to a 64K context length. For our Llama-2-7B-32K experiments using a **25% retrieval head ratio**, the total KV budget under a 32K context length is:
>
> $$\text{Budget Ratio} = \frac{(32K\times 0.25) + (320 \times 0.25)}{32K} = 0.2525$$
>
> This budget is still clearly smaller than the **30%** budget used in RazorAttention’s experiments. Despite this, our **25% ratio results show no degradation**, whereas RazorAttention’s **30% ratio** results exhibit performance drops. This underscores DuoAttention's stronger efficiency and robustness under smaller budget constraints.
>
> We will clarify these points further in the future revision of our manuscript to ensure transparency and address any potential ambiguities.
>
> ------
>
> ### **5. Novelty and Gating-Related Methods**
>
> We appreciate your feedback on DuoAttention’s novelty. While its optimization-based approach marks a significant advance, we acknowledge its connection to prior gating-related works, such as sparsely gated mixture-of-experts (MoE). We will revise the manuscript to better contextualize DuoAttention within this lineage and adjust novelty claims accordingly.
>
> ------
>
> ### **6. Presentation Suggestions**
>
> We agree that Figure 7 can be improved. We will include a table summarizing key LongBench results for better clarity and usability.
>
> ------
>
> ### **7. Query-Dependency of SnapKV-Like Methods**
>
> We concur that query-location sensitivity is a design limitation of SnapKV-like methods. DuoAttention’s query-position robustness offers a clear advantage, which we will further validate with datasets like **SharedContextBench** in future work.
>
> ------
>
> ### **8. Compression Granularity**
>
> DuoAttention delivers practical gains by achieving near-full KV performance at 50% budget, with breaking points illustrated in Figure 7 for various tasks. We will explore these limits further will guide future optimizations.
>
> ------
>
> ### **Final Remarks**
>
> Thank you again for your valuable insights and constructive feedback. We are committed to addressing your suggestions in our revisions and future work.
>
> Best regards,
>
> Authors

---

### Author Response · Authors · 2024-11-26
**General Rebuttal Summary and New Experiments**

We sincerely thank all reviewers for their thoughtful feedback, constructive suggestions, and valuable insights. In this response, we have worked to address each concern comprehensively, conducting new experiments and incorporating clarifications in the manuscript to ensure our contributions are presented accurately and thoroughly. Below is a summary of our key updates and additional findings:

------

### **1. Expanded Comparisons with Baselines**

We conducted new experiments comparing DuoAttention with state-of-the-art KV cache compression methods (SnapKV, PyramidKV) and pre-filling acceleration methods (MInference). These results, detailed in Appendix Sections A.6 and A.7, highlight DuoAttention’s robustness and versatility:

- **KV Cache Compression Methods (SnapKV, PyramidKV)**:
  - DuoAttention outperforms or matches these baselines under equivalent KV cache budget constraints, especially in scenarios involving multi-turn dialogues or queries positioned earlier in the context.
  - SnapKV and PyramidKV rely on assumptions about query positions, which limit their generalizability. DuoAttention operates without such assumptions, ensuring applicability across diverse tasks and contexts.
- **Pre-filling Acceleration Methods (MInference)**:
  - MInference, while orthogonal to KV cache compression, complements DuoAttention by further accelerating the pre-filling stage.
  - Combined with DuoAttention, MInference achieves significant reductions in KV cache size and decoding overhead while maintaining accuracy.

------

### **2. NIAH Benchmark Clarifications**

In response to concerns about experimental setup transparency, we updated Appendix Section A.10 to provide detailed descriptions of the NIAH benchmark, which ensures reproducibility and clarity of our experimental setup.

------

### **3. New Results on the RULER Benchmark**

We extended our evaluation to include the RULER benchmark, a dataset designed to rigorously test long-context capabilities. The results, presented in Appendix Section A.8, demonstrate DuoAttention’s effectiveness:

- **Comparable Accuracy**: DuoAttention achieves accuracy scores comparable to full attention across all context lengths, with an average performance increase of 0.05%.
- **Robustness Across Tasks**: DuoAttention consistently delivers strong performance across diverse long-context tasks, further validating its generalizability and utility.

------

### **4. Combination with KV Quantization**

To address concerns about combining DuoAttention with KV quantization techniques, we conducted additional experiments using INT4 KV Pre-rope quantization (KIVI). These results, detailed in Appendix Section A.9, show that the combined approach introduces a negligible performance drop (0.016 on average) compared to quantization alone.

------

### **5. Detailed Analysis of Performance**

We provided a deeper analysis of DuoAttention’s performance across tasks (e.g., Figure 7 in the manuscript). Key findings include:

- **Task-Specific Patterns**: DuoAttention achieves higher compression ratios for the majority of tasks (e.g., DuReader, MultiNews) while requiring more retrieval heads for some tasks (e.g., TriviaQA). Variations are attributed to differences in task characteristics, languages, and context sizes.
- **Universal Strengths**: Despite these variations, DuoAttention consistently outperforms baselines and guarantees performance comparable to full attention at a 50% KV cache budget for GQA models and 25% for MHA models.

------

### **6. Scalability in Multi-GPU Settings**

We addressed concerns about DuoAttention’s scalability in tensor-parallel configurations:

- **Load Balancing**: Retrieval heads can be distributed evenly across GPUs by aligning their number with the number of tensor-parallel GPUs.
- **Parallelism Compatibility**: DuoAttention supports sequence and pipeline parallelism, ensuring efficient scaling in multi-GPU environments.

------

### **7. General Robustness**

DuoAttention’s design avoids reliance on positional assumptions, enabling it to handle diverse real-world scenarios effectively. Unlike baseline methods such as SnapKV, which focus on end-positioned queries, DuoAttention excels in tasks with varied query positions, such as multi-turn dialogues.

------

### **8. Manuscript Improvements**

Based on reviewer feedback, we revised the manuscript to improve clarity and address inaccuracies:

- Corrected typos and clarified mathematical notations (e.g., Equation 2).
- Updated statements regarding baselines.
- Enhanced explanations of experimental setups, methodologies, and results.

------

We hope these updates and clarifications address your concerns and highlight DuoAttention's contributions and strengths. Please let us know if you have further questions or require additional information, as we are committed to providing any needed clarifications. Thank you for your valuable feedback and thoughtful review!

---

### Author Response · Authors · 2024-11-26
**General Response to Reviewers and ACs**

We sincerely appreciate all reviewers' efforts for the insightful and thoughtful comments. We are glad that the reviewers recognized the following strengths:

1. **Performance and Optimization**: DuoAttention achieves impressive efficiency improvements, significantly optimizing memory usage and inference speed, as highlighted by Reviewer 4AyC. The thorough evaluation of efficiency optimizations further underscores its practical impact.

2. **Novelty and Methodology**: The optimization-based head selection approach, providing a fresh perspective on attention head pattern matching, is appreciated for its unique methodology (Reviewer 4AyC). The novel classification of attention heads into retrieval and streaming types, as outlined in the paper, adds a significant contribution to the field (Reviewer oer2 and Reviewer n7ZA).

3. **Experimental Rigor**: Comprehensive ablation studies and evaluations on various benchmarks demonstrate DuoAttention's robust performance, as noted by Reviewers oer2 and JVPb. The experiments cover state-of-the-art comparisons and present strong results on long-context benchmarks.

4. **Presentation and Clarity**: Our paper is well-structured, with clear writing and motivation. Reviewers 4AyC, oer2, and JVPb found the claims and presentation easy to follow, supported by well-illustrated figures and transparent methods.

These acknowledgments highlight the value of DuoAttention in addressing critical challenges in long-context LLM inference. We are committed to refining our work based on the constructive feedback provided.

---

### Author Response · Authors · 2024-11-28
**Comparison with SnapKV, PyramidKV, and AdaKV on LongBench**

Dear Reviewers,

Thank you for your thoughtful comments and the opportunity to provide additional clarification. We have conducted a detailed comparison between **DuoAttention**, **SnapKV**, **PyramidKV**, and **AdaKV** using the Llama-3-8B-Instruct-Gradient-1048k model under the same **50% KV cache budget**. The results are shown below:



| Dataset             | Full  | SnapKV    | PyramidKV | AdaKV     | DuoAttention |
| ------------------- | ----- | --------- | --------- | --------- | ------------ |
| Average             | 40.08 | 38.47     | 38.39     | 38.67     | **40.21**    |
| 2WikiMQA            | 28.78 | 29.00     | 28.12     | 28.97     | **29.08**    |
| DuReader (zh)       | 30.41 | 24.04     | 26.63     | 22.65     | **29.31**    |
| GovReport           | 34.23 | 26.84     | 27.59     | 24.22     | **32.72**    |
| HotpotQA            | 40.37 | 40.86     | 41.56     | 40.23     | **41.63**    |
| LCC                 | 38.19 | 38.83     | 37.59     | 39.67     | **44.16**    |
| LSHT (zh)           | 38.00 | 38.00     | **38.50** | 36.50     | 30.00        |
| MultiNews           | 27.73 | 22.84     | 22.93     | 21.81     | **27.72**    |
| MultiFieldQA-en     | 52.62 | 51.96     | 52.54     | **52.99** | 51.44        |
| MultiFieldQA-zh     | 50.58 | 50.74     | 49.85     | 50.59     | **52.40**    |
| Musique             | 24.22 | **24.86** | 24.63     | 24.68     | 24.65        |
| NarrativeQA         | 26.56 | 26.63     | 26.17     | **27.36** | 24.54        |
| Passage Count       | 1.00  | **1.00**  | 1.00      | 1.00      | 0.00         |
| PassageRetrieval-en | 81.00 | 80.50     | 80.00     | 80.50     | **87.00**    |
| PassageRetrieval-zh | 62.15 | 58.53     | 54.56     | 61.92     | **62.15**    |
| Qasper              | 29.21 | 26.00     | 23.63     | **27.02** | 26.93        |
| QMSum               | 24.52 | **24.90** | 24.45     | 24.65     | 24.20        |
| RepoBench-P         | 38.94 | 38.20     | 37.48     | 38.50     | **46.12**    |
| SAMSum              | 42.51 | 40.90     | 40.90     | 41.38     | **41.83**    |
| TREC                | 71.50 | 66.00     | 70.00     | 71.00     | **71.00**    |
| TriviaQA            | 87.70 | **87.30** | 87.20     | 86.80     | 87.14        |
| VCSUM (zh)          | 11.37 | 9.91      | **10.80** | 9.62      | 10.46        |



Key Observations:

1. **Best Accuracy Across Benchmarks**: DuoAttention achieved the highest average score (**40.21**) among all methods. It consistently outperformed SnapKV, PyramidKV, and AdaKV across the majority of tasks.
2. **Robustness to Query Positioning**: Building upon the studies in **Appendix Section 6** and **Appendix Section 12**, we observed that SnapKV-style methods (including PyramidKV and AdaKV) rely heavily on an observation window of tokens to identify important KV cache entries. These methods are sensitive to query positioning and struggle in scenarios involving continuous pre-filling or multi-round dialogue, where queries are not always located at the end of the context. DuoAttention does not depend on this heuristic, enabling superior performance in a broader range of applications.
3. **Generality and Applicability**: DuoAttention excels across various tasks and demonstrates resilience to real-world complexities like various query placement, continuous pre-filling, and multi-round dialogue, unlike other KV cache compression techniques.

We believe these comprehensive experiments address your concerns and provide strong evidence supporting the accuracy, generality, and significance of DuoAttention. Please feel free to reach out with additional questions. If our responses meet your expectations, we would greatly appreciate your consideration of improving the paper’s rating.

Thank you for your time and thoughtful feedback.

Best regards,

Authors

---

### Meta-Review · Area_Chair_3dGc · 2024-12-21

**Metareview:**

This paper presents DuoAttention, a method designed to make long-context inference in large language models (LLMs) more efficient. The approach reduces memory use and processing time by focusing full key-value caching on a subset of attention heads—referred to as retrieval heads—while applying a streamlined cache strategy to other heads. The authors propose an optimization-based method for identifying these critical attention heads, achieving significant improvements in efficiency without compromising performance on long-context tasks. The experiments demonstrate up to 2.55x memory savings and 2.18x faster processing for specific models.

The paper is well-written and addresses an important challenge in deploying LLMs with long-context capabilities. Its strengths include a clear motivation for the proposed method, strong empirical results, and a thoughtful evaluation across multiple benchmarks. However, some limitations remain. The work could benefit from more extensive testing on real-world datasets and a broader range of baseline comparisons. Additionally, while the optimization strategy is effective, the paper would be stronger if it provided deeper insights into how the method generalizes across diverse tasks.

This paper is recommended for acceptance due to its potential to advance the efficiency of LLMs in long-context scenarios. The strong experimental results and practical implications outweigh the identified gaps.

**Additional Comments On Reviewer Discussion:**

Reviewers raised questions about novelty, the scope of baseline comparisons, and the real-world applicability of the method. The authors responded by adding comparisons with newer techniques and expanding their experimental evaluations. While some concerns about generalizability and novelty were noted, the authors’ thorough responses and the compelling experimental results make a strong case for the paper’s acceptance.

---

### Decision · Program_Chairs · 2025-01-22

Accept (Poster)